# A hybrid CNN-XLSTM-GRU deep learning model with autoencoder-based feature selection for hypothyroidism diagnosis

**Divya Kesavulu** , **Kannadasan R** *

School of Computer Science and Engineering, Vellore Institute of Technology, Vellore, India

* divya.k2022@vitstudent.ac.in, kannadasan.r@vit.ac.in

## Abstract

Hypothyroidism, caused by reduced thyroid-hormone production, is often difficult to diagnose because its symptoms are non-specific and overlap with other disorders. We developed FusionNet-CXG, a hybrid deep-learning model that couples CNN, eXtended LSTM, and GRU components for hypothyroidism prediction. Experiments were performed on a public dataset (3,772 records; 30 features). Class imbalance was mitigated using SMOTE-NC, and performance was estimated using 10 repetitions of 5-fold stratified cross-validation, yielding 50 fold-level evaluations to obtain stable results. Across folds, FusionNet-CXG achieved 0.9394 mean accuracy, F1 = 0.906, and AUC-ROC = 0.94, and it exceeded CNN+LSTM and CNN+BiLSTM baselines under the same protocol. The results indicate that combining local feature extraction with recurrent modelling is effective for this task. To support interpretability, SHAP was used to quantify feature influence on the predictions. Future work will focus on external validation using more diverse real-world cohorts and incorporating temporal clinical data to improve clinical relevance.

## 1. Introduction

Thyroid disorders impose a considerable public health burden in India, impacting nearly 42 million people. Data from the National Family Health Survey indicate that self-reported goitre or thyroid conditions rose from 2.2% in 2015–2016 to 2.9% in 2019–2021. Hypothyroidism is also widely reported, with about one in ten individuals in India affected. Women are impacted far more than men, with thyroid problems reported to be 6–10 times more common among females [1]. In addition, around 10%–20% of women above 40 years are affected. Even though thyroid disorders are common, awareness and treatment remain inadequate. This points to the need for stronger public health efforts focused on early identification, community education, and effective management of thyroid disorders [1].

Thyroid diseases encompass a range of medical conditions that disrupt the normal functioning of the thyroid gland, an endocrine organ located in the neck region [2,3].

**Data availability statement:** https://www.kaggle.com/datasets/kumar012/hypothyroid.

**Funding:** The author(s) received no specific funding for this work.

**Competing interests:** The authors have declared that no competing interests exist.

The thyroid gland governs metabolism, energy generation, growth, and development [4]. Common thyroid problems include hypothyroidism. Both functional and neoplastic thyroid diseases are common, and they are identified manually [4,5]. To identify thyroid functional illness, such as hypothyroidism [6] and hyperthyroidism, a thyroid function examination is necessary. Triiodothyronine (T3) and thyroxine (T4) are the two primary hormones that the thyroid glands produce [7,8]. In addition to T3 and T4, thyroid-stimulating hormones (TSH) and free T3 (FT3) and free T4 (FT4) make up the thyroid function test used to identify hypothyroidism and hyperthyroidism [9,10].

Hypothyroidism, a common medical disorder resulting from inadequate thyroid hormone production, impacts a significant portion of the global population [11,12]. Its manifestations, including exhaustion, increased body mass, and reduced mental acuity, frequently resemble those of other ailments, complicating prompt and precise identification [13,14]. Conventional diagnostic methods, which rely on blood analyses to measure thyroid-stimulating hormone (TSH) and thyroid hormones (T3, T4), are generally effective but often struggle to identify subtle indicators of early-stage disease [15,16]. This highlights the necessity for sophisticated diagnostic tools that harness modern computational approaches [17,18] like more advanced and intelligent diagnostic solutions.

Artificial intelligence (AI) is increasingly reshaping medical diagnostics by enabling models to uncover complex relationships in clinical data that traditional approaches often miss. In this context, machine learning (ML) and deep learning (DL) methods have gained clear traction for thyroid disease prediction. In practice, CNNs are often the first choice for imaging-based diagnosis, whereas RNN families such as LSTM and GRU are better suited to hormone readings collected over time, where the order of measurements matters. Despite these advances, current methods still face three main limitations: (i) class imbalance in available datasets, which biases predictions toward the dominant classes; (ii) limited feature representation, where conventional dimensionality reduction and manually engineered features often fail to preserve clinically important information; and (iii) weak modeling of spatio-temporal dependencies, since CNN-based models mainly learn spatial patterns and RNN-based models focus on temporal dynamics, leaving spatial relationships underexplored.

The motivation for this study comes from two angles: clinical need and computational opportunity. In clinical practice, there is still a strong need for dependable AI-assisted tools that help detect hypothyroidism early, lower the risk of misdiagnosis, and support better patient outcomes. On the modeling side, many existing approaches are constrained by imbalanced datasets, noisy or redundant feature spaces, and limited ability to learn spatial and temporal patterns together in thyroid-related data. To overcome these gaps, we introduce FusionNet-CXG, a hybrid deep learning framework that combines CNNs, eXtended LSTM (XLSTM), and GRUs so that spatial patterns, long-range dependencies, and short-term fluctuations can be learned in a unified manner. We integrate SMOTE-NC to handle class imbalance in datasets with both categorical and continuous variables, and we apply an autoencoder-based feature extractor to compress the feature space while preserving clinically informative signals. Finally, performance alone is not sufficient for clinical

deployment: clinicians need models that are accurate and transparent about the factors driving their decisions. Therefore, SHAP (SHapley Additive exPlanations) is used to explain individual predictions by showing how each clinical feature contributes, which improves interpretability and supports trust in the framework.

This study contributes the following:

1. A hybrid deep-learning architecture, FusionNet-CXG, is introduced by combining CNN, XLSTM, and GRU modules so that both feature interactions and sequential patterns in hypothyroid-related clinical data can be learned effectively.

2. An autoencoder-based feature extractor is incorporated to clean noisy inputs, compress redundant information, and produce compact representations that improve generalization.

3. SMOTE-NC is used on the mixed-type clinical variables to balance the classes and limit bias during training.

4. Compared with strong baselines such as CNN+LSTM and CNN+BiLSTM, the proposed approach achieves 92.84% accuracy, a 0.91 F1-score, and a 0.94 AUC-ROC.

5. Robustness is further examined using statistical significance tests and gender-wise performance analysis, which supports the reliability of the model in a clinical setting.

6. Interpretability is strengthened using SHAP, which highlights the most influential clinical features and makes the predictions easier to justify in a medical context.

Overall, the work addresses practical clinical needs while closing technical gaps seen in earlier studies, and it provides a clear base for future validation on real-world, longitudinal datasets and explainable AI-enabled healthcare systems.

The novelty of FusionNet-CXG comes from bringing three complementary components into one pipeline: SMOTE-NC for balancing imbalanced mixed-type clinical data, an autoencoder for reducing noise and redundancy in the feature space, and a CNN–XLSTM–GRU backbone that captures spatial relationships together with short- and long-term temporal dependencies. Compared with earlier approaches that rely on CNNs or recurrent models alone, this integrated design improves generalization and supports more consistent hypothyroidism prediction.

## 2. Related work

Machine learning (ML) and deep learning (DL) have advanced thyroid disease diagnosis considerably, and many published models report strong accuracy [19,20]. Even so, several recurring issues remain, including class imbalance, weak feature representation, and limited handling of temporal information. In the following discussion, we group recent studies by their methodological approaches and highlight the key limitations reported in each line of work.

### 2.1. CNN-based models for thyroid disease prediction

Many studies have applied Convolutional Neural Networks (CNNs) to thyroid disease classification. In [10], the authors designed a multi-channel CNN to detect thyroid cancer from medical images and obtained high classification accuracy. In a related study, [2] used CNN-based analysis of SPECT scans to identify thyroid disorders and achieved encouraging performance. However, most CNN-based methods primarily learn spatial patterns and offer limited support for temporal modelling. This becomes a constraint for hypothyroidism prediction, where the sequential behaviour of hormone levels plays a critical role.

### 2.2. LSTM and GRU-based models for thyroid prediction

For sequence-style clinical data, RNNs remain the standard baseline, with LSTM and GRU being the most used variants. In [21], the residual LSTM design was improved via the Giant Trevally Optimizer (GTO) for thyroid disorder prediction, and improved performance was reported. In [22], VGG-19 handled feature extraction in a CNN–LSTM pipeline,

delivering 98.8% accuracy. Despite their effectiveness on sequential inputs, these models often have difficulty preserving long-term dependencies and generally do not capture spatial information when structured feature relationships exist in the data.

## 2.3. Hybrid deep learning approaches

Hybrid designs have been suggested mainly because a single model rarely covers everything well. Along these lines, [23] proposed DEL-Thyroid, an ensemble that combines LSTM, GRU, and BiLSTM, and reported 96% accuracy. Another related study, [24], examined semi-supervised learning together with the Internet of Everything (IoE) for thyroid disorder prediction and found that it handled imbalanced datasets more effectively. Even with these hybrid directions, the same issues still appear repeatedly, feature selection remains difficult, class imbalance is not fully resolved, and interpretability is often not strong enough for clinical use.

The study [25] proposes an ensemble framework combining CNN, LSTM, and MLP models to diagnose diabetes from clinical data. The CNN extracts spatial representations, the LSTM captures temporal dependencies, and the MLP integrates these features. Their ensemble (e.g., via voting or stacking) outperforms single-model baselines, improving diagnostic accuracy and robustness [26]. introduces a deep stacking ensemble for early heart disease prediction, combining optimized CNN–LSTM and CNN–GRU base learners with an SVM meta-learner. Using RFE for feature selection and evaluating on two heart-disease datasets, the ensemble outperforms traditional ML baselines and individual hybrid [27]. uses ensemble deep learning models combining CNN with BiLSTM/BiGRU variants, showing that these ensembles achieve superior performance over individual models in heart disease classification. An attention-based [28] hybrid CNN–LSTM is applied to cough audio, paired with spectral/data augmentation (e.g., pitch-shift on waveforms and SpecAugment on Mel-spectrograms), yielding ~91% test accuracy and AUC for COVID-19 diagnosis; ablations show both attention and augmentation drive the gains [29]. propose an ensemble deep-learning approach (called ALZ-IS) for Alzheimer's disease identification. Their system combines multiple neural architectures (e.g., CNN variants) into a hybrid ensemble, leveraging complementary strengths to improve robustness and outperform individual models in Alzheimer's identification. They also provide an online interface and argue that the method improves accuracy, generalizability, and day-to-day diagnostic usefulness for Alzheimer's identification.

More broadly, steady advances in machine learning and deep learning have expanded their role across healthcare, supporting the move toward AI-assisted clinical decision making. For example, the federated learning–based system in [30] addresses automated diabetic foot ulcer diagnosis using consumer-grade devices, showing that privacy-preserving, distributed training can be implemented in realistic medical workflows. The review in [31] surveys drug–target interaction prediction and outlines how machine learning can speed up drug discovery and help reveal new therapeutic opportunities. HCNNet, a hybrid convolutional network, has been used to detect ischemia in diabetic foot ulcers [32], suggesting that hybrid models can be effective for clinical image analysis. Separately, growth-based modelling has been applied to track and predict COVID-19 trends [33], showing how AI can aid epidemiology and public health planning. IoT-based sensing combined with machine learning is increasingly being used to build smart healthcare platforms that can tailor care using data collected in real time [34]. In parallel, physiological-signal–based methods are being investigated for early cognitive-decline detection, targeting changes that may precede overt symptoms [35].

## 2.4. Feature selection and class imbalance handling

Several studies have focused on addressing class imbalance and feature selection. In [36], the Blunge Calibration Classification Model (BCCM) combined Kernel SVM, K-Nearest Neighbors (KNN), and Ridge estimators, and reported an accuracy of 99.7%. Another study [37] applied synthetic oversampling techniques for thyroid disease diagnosis. Even so, these conventional ML approaches often rely on manual feature engineering, which may be less suitable for large-scale clinical datasets.

Overall, these works highlight how ML and DL methods can be applied to diverse healthcare problems, including imaging-based diagnosis, disease forecasting, and intelligent healthcare systems. Building on these developments, our proposed FusionNet-CXG framework extends hybrid deep learning to hypothyroidism prediction, with a specific focus on mitigating dataset imbalance, improving feature representation, and strengthening spatial–temporal modeling.

### 2.5. How FusionNet-CXG improves upon previous models

Current models still struggle in three areas:

- Temporal learning: CNN-based models are limited for sequential dependencies.

- Feature selection: many methods rely on conventional feature engineering, which can hurt generalization.

- Imbalance: standard oversampling is common in prior work and can cause overfitting.

FusionNet-CXG addresses these issues by using CNN (spatial features), XLSTM (long-term dependencies), and GRU (short-term trends) to strengthen sequential learning. SMOTE-NC is used to balance class distribution, and an autoencoder supports feature selection by reducing dimensionality and noise. This integrated design improves predictive accuracy.

## 3. Materials and methods

Deep learning has grown to be a hot topic in machine learning over the last several years. Unquestionably, machine learning has improved the diagnostic accuracy rates for identifying thyroid disorders [38]. Deep learning, on the other hand, increases the accuracy and efficiency of diagnosis by automatically selecting characteristics from inputs [39]. Deep neural networks are a novel approach to categorisation, and many researchers have used them to identify a wide range of illnesses, including lung cancer, breast cancer [40], melanoma [41], and macular oedema.

Fig 1 presents the methodology suggested to be used in the study in a sequential format. The study utilized an open-access dataset of hypothyroid that contained 3,772 observations and 30 variables [42]. Because of the nature of the

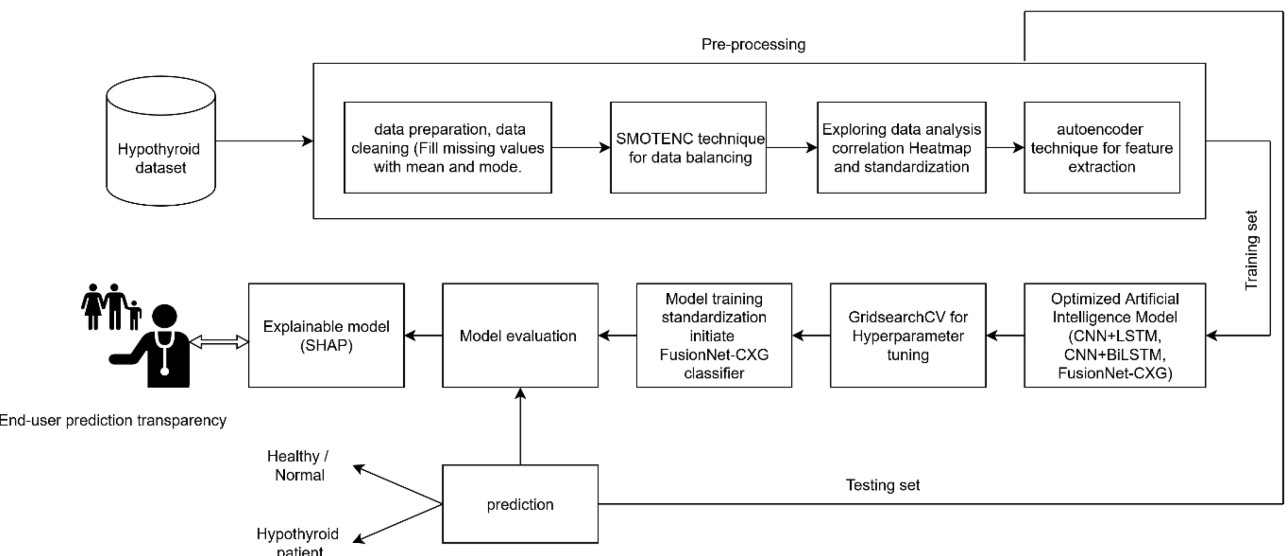

**Fig 1. Workflow of the proposed FusionNet-CXG framework for hypothyroidism prediction.** The pipeline integrates preprocessing, class balancing, feature extraction, and a hybrid CNN–XLSTM–GRU classifier.

imbalance in the dataset, class balancing metrics were incorporated in the preprocessing stage. The model performance was evaluated using 5-fold stratified cross-validation using 10 iterations. In both iterations, a fraction of 4 folds (around 80% of the data set) was used as a training set, and another fraction (around 20% of the data set) was used as an evaluation set. The training segment involved running such procedures as preprocessing, class balancing, feature extraction, and model fitting and the implementation of an internal validation split and early stopping in the training process. In order to provide a complete reporting of the performance metrics, the aggregation of the results was done over all the cross-validation iterations. In order to make statistical comparison between FusionNet-CXG and the baseline models, the accuracy values over the five folds in each iteration of the experiment were averaged to produce a single mean accuracy of each model in an iteration. This procedure brought about 10 paired observations of repetition level with each model comparison. The normality of the paired differences was conducted before using the paired t-test. This methodological technique blocked the statistically independent observations of the results of the same fold-level.

### 3.1. Dataset details

A publicly available hypothyroidism dataset from Kaggle [29] is used in this study. It contains 3,772 patient records described by 30 clinical attributes. These attributes cover demographic information, medical history, treatment-related details, and thyroid hormone test results, supporting a binary classification problem (hypothyroid vs. non-hypothyroid). The main characteristics are:

• Demographic Details – Age, gender.

• Medical History – Prior thyroid therapies, current pregnancy status, history of thyroid surgery, presence of tumors.

• Thyroid Function Assessments (TFAs) – T3, TSH, Total TT4 (T4), Free T4 Index (FTI), T4 Uptake (T4U), and Thyroxine-Binding Globulin (TBG).

• Target Variable – A binary label representing hypothyroid condition: P (Positive) or N (Negative).

Nonetheless, this dataset, first gathered in 1987, presents limitations since medical diagnostic standards and methods for thyroid testing have advanced over the years. Additionally, class imbalance presents a significant challenge, as there are far more non-hypothyroid instances than hypothyroid ones, potentially skewing model predictions in favour of the majority class.

Fig 2 shows the count distributions of the raw TSH, TT4, T3, and FTI before any preprocessing, and the hormones do not follow a single pattern. TSH is heavily right-skewed: most patients have low readings, with a smaller group extending into the high range often linked with hypothyroidism. T3 leans the same way, mostly on the lower side. TT4 is different, with a more even spread closer to a bell curve and wider differences between patients. FTI shows a sharp central peak with only moderate spread. Taken together, these mixed patterns make clear that normalization and careful outlier handling are necessary before building prediction models.

The dataset was also uneven in terms of gender. Fig 3 shows a clear gender skew in the dataset, with nearly 70% of cases from women and about 30% from men. This pattern is in line with clinical reports that thyroid disorders are more prevalent among women. Noting this imbalance is important because, while it suggests the data reflect real-world prevalence, it also indicates that mitigation steps (such as class balancing) are needed to reduce the risk of the model learning biased gender-driven cues.

### 3.2. Data preprocessing

To improve model performance, the dataset was processed using the following steps:

**3.2.1. Addressing missing data.** A preprocessing stage was applied to handle missing values, with the goal of maintaining data quality and supporting stable model performance. In this dataset, missing entries appear as the placeholder "?". These entries are imputed according to attribute type. For numerical variables, mean imputation is used

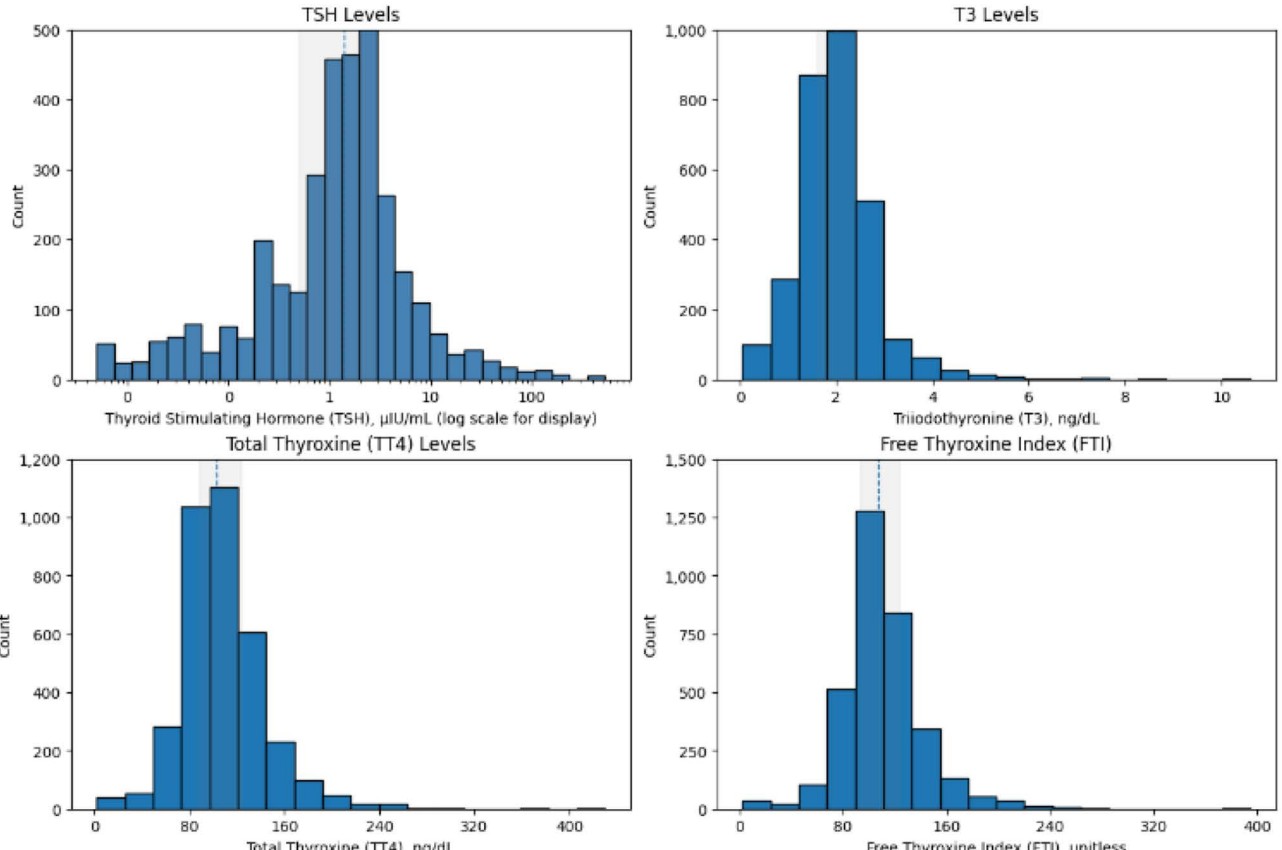

**Fig 2. Distribution of thyroid-related hormones (TSH, TT4, T3, and FTI) in the dataset.**

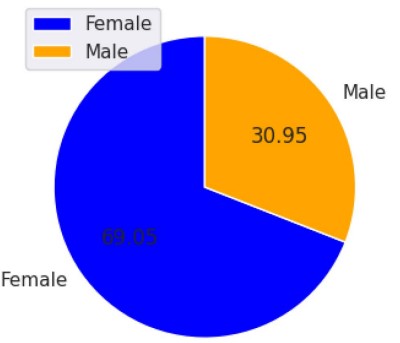

Thyroid disease affecting rate on male vs female

**Fig 3. Gender distribution of patients in the dataset.**

so that overall statistics are preserved without causing large shifts in the data distribution. For categorical variables, mode imputation is applied to keep category frequencies consistent across the dataset. For certain binary attributes, missing entries are also assigned to an additional category so that potentially informative missingness patterns are retained.

To examine the effectiveness of the imputation procedure and to visualize missing-value patterns, a heat map (Fig 6) is generated, summarizing the distribution of missing data and supporting the preprocessing strategy adopted.

**3.2.2. Encoding categorical variables.** To enable machine learning processing, the categorical variables in the dataset are converted into numerical formats by employing suitable encoding techniques. Binary encoding converts binary categorical variables such as gender (Male/Female) by assigning the values 0 and 1. One-hot encoding is utilized for categorical data featuring multiple unique categories like various treatment methods to generate separate binary columns for each category. This transformation ensures that categorical features remain interpretable while being represented suitably for computational models. These encoding techniques assist the dataset in preserving its structural integrity, thus allowing the model to identify patterns and relationships within the data.

**3.2.3. Feature normalisation.** To train and assess the model, 5-fold stratified cross-validation repeated 10 times was used. In each split, approximately 80% of the data (four folds) was used for training and approximately 20% of the data (one fold) was used for fold-wise evaluation. An internal validation split was further used within the training partition for early stopping.

Normalization was applied in two stages:

1. Apply Min–Max scaling to the range [0,1] prior to autoencoder training to synchronise data with the sigmoid activation utilised in reconstruction.

2. Application of Z-score standardisation post-dimensionality reduction to guarantee uniform contribution of latent features for CNN–XLSTM–GRU classification.

Formally, Z-score scaling is defined as:

$$X_{normalized} = \frac{X - \mu}{\sigma}$$

(1)

Where x is the raw data value before normalisation. $\mu$ is mean and $\sigma$ is the standard deviation.

## 3.3. Class balancing

In the training sections of the repeated stratified cross-validation step, the data were highly skewed against the negative data, where there were much fewer cases of hypothyroid as compared to the non-hypothyroid. To solve this imbalance, various resampling techniques were analyzed such as Random Oversampling, Random Under-sampling, ADASYN, Borderline-SMOTE, and SMOTE-NC. Because the distribution of classes was still very skewed, balancing is considered a preprocessing action. Out of the experimented methods, SMOTE-NC exhibited the most reliable performance; thus, it was used in the final experiments, which was also compared to the other balancing methods.

Fig 4 shows the original class distribution alongside the effect of each balancing strategy. All oversampling-based approaches, including Random Oversampling, ADASYN, Borderline-SMOTE, and SMOTE-NC, produced nearly balanced datasets. By contrast, Random Under-sampling reduced both classes to just 204 samples, resulting in a major loss of information. We ultimately chose SMOTE-NC for model development because it generates realistic synthetic samples in datasets with both categorical and numerical variables, a common feature of clinical data.

## 3.4. Feature extraction with autoencoder

Feature selection is an essential step for reducing dimensionality and strengthening model performance. In this work, an autoencoder is used as a deep learning–based feature extraction method that compresses the original variables into a latent representation while retaining clinically relevant diagnostic information, thereby avoiding manual feature-selection procedures [43]. In this framework, the encoder draws the input variables into a more compact latent space,

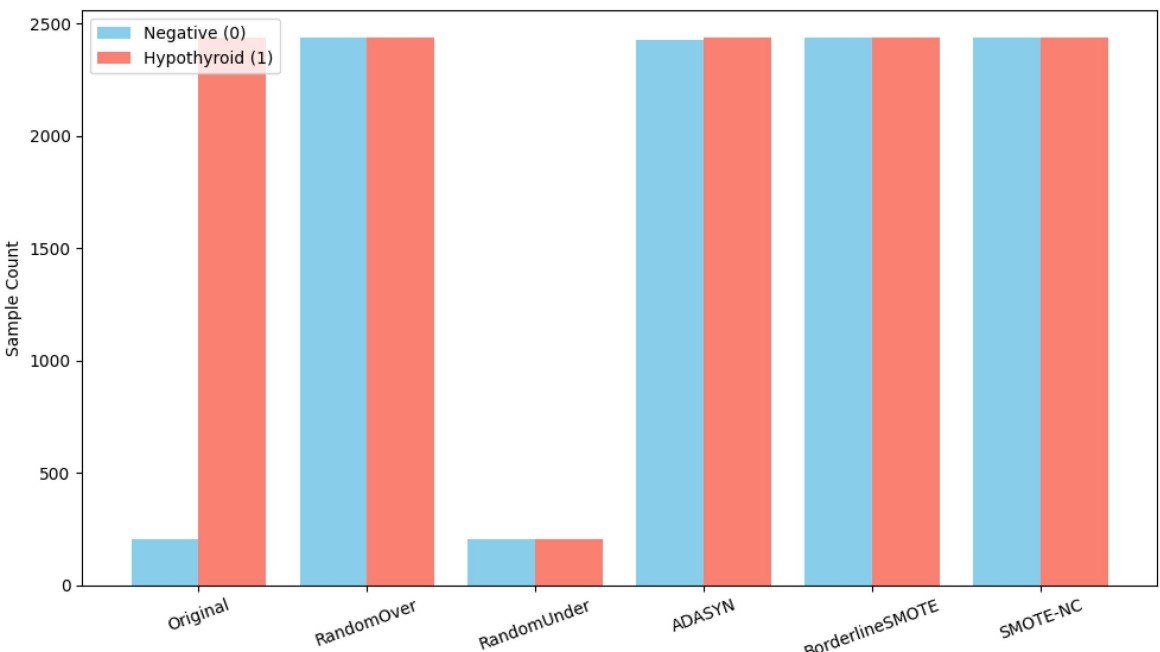

**Fig 4. Comparison of class balancing techniques.**

and the decoder reconstructs the data from this compressed form so that important patterns are preserved. Training is guided by the reconstruction loss, which encourages the latent space to keep the features that are most useful for subsequent classification. By relying on this learned representation, the approach reduces the subjectivity of manual feature engineering and helps protect meaningful clinical details, ultimately improving computational efficiency and predictive accuracy [44].

After applying the autoencoder, the original set of 30 features was reduced to 17, retaining the variables most relevant to the classification task. This reduction supports stronger generalization and lowers the risk of overfitting by removing noisy or redundant attributes [44]. The correspondence between the original and encoded features, shown in Fig 10, illustrates the effectiveness of this dimensionality-reduction technique [45].

Fig 5 displays the scatter plot matrix for selected thyroid-related variables, including age, TSH, T3, TT4, T4U, and FTI. Several clear relationships are visible. TSH shows a strong inverse association with most other hormone levels, reflecting the well-known physiological pattern where elevated TSH is linked with reduced thyroid hormones. T3, TT4, and FTI moved in the same direction, showing positive correlations that indicate these hormones often rise together. Age, on the other hand, had a weak negative association with TSH, with older patients generally recording lower TSH values. The scatter plot makes these trends visible and also shows which variables overlap in their information and which provide distinct signals that can strengthen the predictive model.

Fig 6 shows the correlation heatmap, providing a single snapshot of how the variables relate to one another. Darker regions reflect stronger positive links, most clearly the grouping of T3, TT4, and FTI, while lighter shades point to weaker or negative associations, particularly between TSH and the other thyroid measures. The categorical variables show only minor connections with the numerical ones. Overall, the heatmap makes it clear that some features overlap in the information they provide, whereas others contribute more distinct patterns. This distinction underscores the value of reducing redundant features before building the model.

**Fig 5. Relative contribution of key features.**

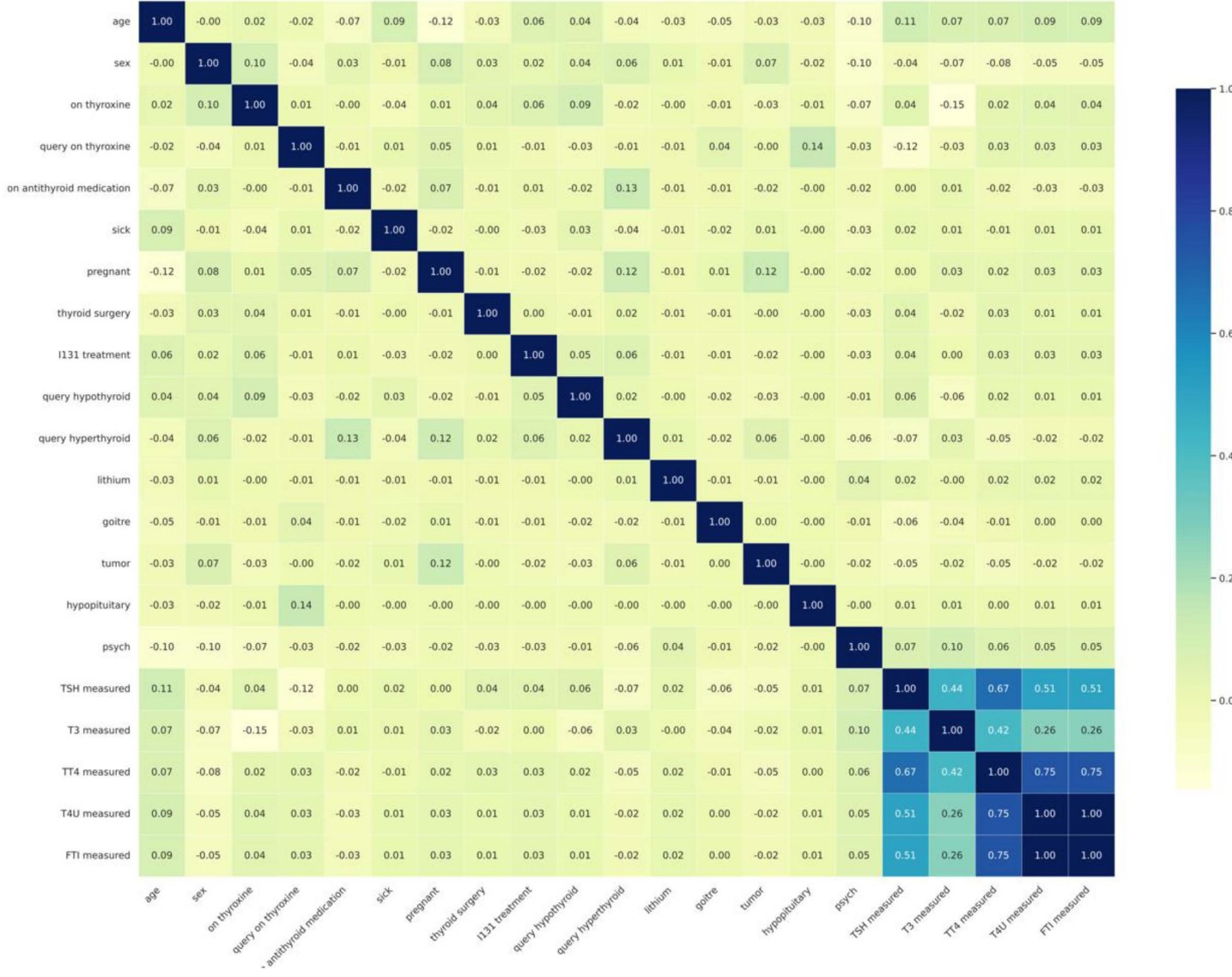

**Fig 6. Heatmap of pairwise feature correlations.**

### 3.5. Proposed hybrid model (FusionNet-CXG)

FusionNet-CXG is designed as a hybrid deep learning framework for predicting hypo-thyroidism. Instead of relying on a single architecture, it brings together several modules, with each component addressing limitations noted in earlier methods. In combination, these modules enable the network to learn both spatial relationships and temporal behaviour in thyroid-related data, while also providing mechanisms to handle class imbalance and redundant clinical features. For clarity, the overall workflow can be viewed as proceeding through five broad stages.

Data preparation forms the first stage of the pipeline. Missing entries in the clinical variables are filled using imputation rules appropriate to each variable type, and categorical attributes are translated into numerical codes before model training. Because the original dataset exhibits a skewed class distribution, SMOTE-NC is applied to generate additional

minority instances and produce a more balanced sample. The processed data are then passed through an autoencoder that compresses the 30 original attributes into 17 latent features, reducing noise while preserving information relevant to diagnosis. In the following stage, convolutional layers act on this encoded representation to capture local spatial dependencies among the hormone-related variables.

After spatial features have been extracted, the focus shifts to temporal dynamics in the data. Long-range dependencies are modelled using an extended LSTM (XLSTM), which enables the network to associate patterns that recur over longer sequences. In addition, GRU units are employed to represent short-term fluctuations. Compared with standard LSTMs, GRUs are more compact and computationally efficient, making them suitable for tracking rapid variations in the signal and providing a practical complement to the XLSTM.

Sections 3.3 and 3.4 have already detailed how the data are pre-processed and compressed with an autoencoder to handle imbalance and reduce redundancy. On top of these prepared inputs, the CNN–XLSTM–GRU network acts as the main predictive engine of FusionNet-CXG. By combining these elements, the framework tackles three common shortcomings in earlier work: weak temporal modeling, heavy dependence on hand-crafted features, and errors introduced by skewed class distributions. The overall pipeline is shown in Fig 1.

### 3.6. Hybrid deep learning architecture

**3.6.1. CNN for spatial feature extraction.** CNN is utilized as the initial processing layer to capture spatial relationships from organized input features [46]. A 1D convolutional layer utilizes kernel filters to extract local feature representations, mathematically defined as:

The convolution procedure for each output feature map F is expressed as

$$F(p, q) = \sum_i \sum_j X(p + i, q + j) * K(i, j)$$

(2)

Where in the eq 2, X is the input feature matrix, including patient data across several attributes such as TSH, T3, and TT4 levels. K is the convolutional filter that moves over X. i and j iterate over its dimensions. F(p,q) denotes the value of the convolution output at the coordinates (p,q) inside the feature map. This equation enables CNN layers to concentrate on spatial dependencies, facilitating the acquisition of structural information in medical pictures or interdependencies among features [46].

**3.6.2. XLSTM for long-term dependencies.** LSTM-based architectures demonstrate proficiency in processing sequential data by preserving temporal information via memory cells. The XLSTM layer enhances conventional LSTMs by integrating supplementary memory units [47], thereby augmenting feature retention across extended sequences. The fundamental LSTM equations are:

$$\text{Input gate}: \; i_t = \sigma \left( W_i \times \left[ h_{t-1}, X_t \right] + b_i \right)$$

(3)

$$\text{Forget gate}: \; f_t = \sigma \left( W_f \times \left[ h_{t-1}, X_t \right] + b_f \right)$$

(4)

$$\text{cell state update}: \; \widetilde{C}_t = \tanh(W_c \times \left[ h_{t-1}, X_t \right] + b_c)$$

(5)

$$C_t = f_t \times C_{t-1} + i_t \times \widetilde{C}_t$$

(6)

$$\text{Output gate}: O_t = \sigma(W_o \times [h_{t-1}, X_t] + b_o) \tag{7}$$

$$\text{Hidden state}: h_t = O_t \times \tanh(C_t) \tag{8}$$

Where $X_t$ represents the input at time t. $h_{t-1}$ denotes the preceding hidden state. $C_t$ represents the cell state, encapsulating long-term dependencies. $i_t$, $f_t$, $O_t$ represent input, forget, and output gates. $\widetilde{C}_t$ is the candidate cell state. $W_i, W_f, W_c, W_o$ and $b_i, b_f, b_c, b_o$ are parameters subject to training as weights and biases. $\sigma$ is the sigmoid activation function, $\tanh$ is a hyperbolic tangent.

### 3.6.3. GRU for short-term pattern recognition.
The GRU layer is utilized together with XLSTM to recognize short-term dependencies and enhance computational efficiency [48,49]. The GRU refreshes its hidden state in the following manner:

$$\text{Update gate}: z_t = \sigma(W_z \times [h_{t-1}, X_t] + b_z) \tag{9}$$

$$\text{Reset gate}: r_t = \sigma(W_r \times [h_{t-1}, X_t] + b_r) \tag{10}$$

$$\text{Candidate hidden state}: \widetilde{h}_t = \tanh(W_h \times [r_t * h_{t-1}, X_t] + b_h) \tag{11}$$

$$\text{Hidden state update}: h_t = (1 - z_t) \times h_{t-1} + z_t \times \widetilde{h}_t \tag{12}$$

Where in the eq. 9–12, $z_t$ is the update gate regulates the extent of prior data to be preserved. $r_t$ is the reset gate, regulating the extent of prior information to be discarded. The candidate hidden state $\widetilde{h}_t$ is produced by applying the reset gate to the previous hidden state and integrating it with the current input. Finally, the new hidden state $h_t$ represents the revised hidden state at the time t [47]. $W_z, W_z, W_h$ are trainable weight matrices. $b_z, b_r, b_h$ are trainable biases. $\sigma$ is sigmoid activation function. Tanh denotes hyperbolic tangent activation function.

### 3.6.4. Model integration and output layer.
The output feature maps from CNN, XLSTM, and GRU are combined into one feature vector and fed into a fully connected layer [45,49]. The ultimate classification result is calculated as:

$$H_{hybrid} = [h_{CNN}, h_{XLSTM}, h_{GRU}] \tag{13}$$

Where $h_{CNN}$ denotes the feature vector produced by the CNN block, $h_{XLSTM}$ is the outcome from the XLSTM layer. $h_{GRU}$ is the output from the GRU layer.

Dense layers and final prediction:

The integrated feature vector is subsequently processed through fully connected dense layers using dropout and L2 regularisation, resulting in a final classification layer.

$$output = \sigma(W_{dense} * H_{hybrid} + b_{dense}) \tag{14}$$

Where, $b_{dense}$ and $W_{dense}$ are the bias and weight matrix for the dense layer. $\sigma$ represents the activation function. In this study, the softmax activation function with two output units was employed to simulate the probability distribution for the binary classification task (hypothyroid versus non-hypothyroid) [47].

Pseudocode for proposed FusionNet-CXG model

**Input dataset:** $D \in \mathbb{R}^{n \times m}$ .
Where $n$ represents the number of patient data and $m$ is the number of clinical features (TSH, T3, TT4,…).
**Target variable:** $\mathcal{Y} \in \{0,1\}^n$ (0 = normal, 1 = hypothyroid).
**Output:** Trained FusionNet-CXG model M, cross-validated performance metrics, and SHAP feature attributions.
1. Load Dataset
1.1. Read CSV file into a dataframe.
1.2. Replace placeholder values (?) to NaN.
1.3. Identify the target column and map labels.
1.4. Split X features and y labels.
**2. Data preprocessing**
2.1. Impute missing values using the model for categorical data and the mean for numerical data.

$$x_j = \begin{cases} mean(X_j), & \text{if numerical} \\ mode\ (X_j), & \text{if categorical} \end{cases}$$

2.2. Use one-hot encoding when encoding categorical variables.
2.3. Apply min-max scaling to normalise numerical features.

$$x'_j = \frac{x_j - min(x_j)}{max\ (x_j) - min(x_j)}$$

**3. Cross-validation setup**
3.1. Split dataset using K-fold CV.
3.2. For every fold k, define the training set $T^k$ and validation set $V^k$.
**4. Class balancing**
4.1. Use SMOTE-NC to $T^k$ to create synthetic minority samples.
**5. Autoencoder Feature Extraction**
5.1. Train autoencoder on $T^k$.
5.2. Obtain latent representation $z = f_0(x)$.
5.3. Use $z$ (17-dimensional) and processed features as input for the model.
**6. Hybrid deep model (FusionNet-CXG)**
6.1. CNN component: derive spatial patterns from input features.
6.2. XLSTM branch: Record long-term temporal dependencies.
6.3. GRU: capture of Short-Term Temporal Dynamics.
6.4. Fusion: integrate outputs $[h^{CNN}, h^{XLSTM}, h^{GRU}, \ddagger]$.
6.5. Classification: propagate the fused vector through dense layers to yield output $\hat{y} \in [0,1]$.
**7. Training and optimisation**
7.1. Enhance binary cross-entropy loss using the Adam optimiser.
7.2. Apply dropout and L2 regularisation.
    For epochs=1 to N do
        Train the model on the training set
        Validate the model on the validation set
        Monitor validation loss for early stopping.
    end for
7.3. Use early stopping based on validation loss.
7.4. Optimise hyperparameters using GridSearchCV.
8. Evaluate model on test/validation sets.
9. Preform prediction using FusionNet-CXG.
10. Calculate accuracy, precision, recall, and F1-score on $V^k$.
    Average metrics across folds (mean±std).
**11. Interpretability**
11.1. Apply SHAP to compute feature importance values $\varnothing_j$.
11.2. Prioritise clinical attributes based on their predictive significance.
Return: final FusionNet-CXG classifier M, evaluation metrics, and SHAP explanations.

## 3.7. Model hyperparameters and training configuration

In this approach, we used GridSearchCV to tune the FusionNet-CXG model for hypothyroid diagnosis. The grid covered optimiser type, dropout rate, batch size, and a maximum number of epochs [50]. Within each cross-validation fold, we trained with early stopping (monitoring validation loss and restoring the best weights), so the epoch count served only as an upper bound, and training could stop earlier if performance became constant. GridSearchCV ran k-fold cross-validation, fitting on k−1 folds and evaluating on the held-out fold, which gave a broad view of generalisation and reduced overfitting risk. We chose the best setting from the cross-validated scores and then refit the model on the training data. While random search or Bayesian optimisation can be faster, GridSearchCV lets us examine the space thoroughly and provided a reliable baseline.

The optimal hyperparameters of the proposed model are shown in Table 1. Adam was used as the optimizer for stable convergence, with a dropout rate of 0.5 to control overfitting. A batch size of 128 and 500 training epochs ensured efficient learning, and model robustness was confirmed using 5-fold cross-validation.

Unlike standard CNN+LSTM designs, FusionNet-CXG has an XLSTM layer that maintains long-term dependencies, while GRU improves short-term pattern recognition. In addition, an autoencoder compresses input characteristics to improve efficiency while preserving information. Tables 2, 3 depicts the overall model architecture.

**Table 1. Optimal Hyperparameter Settings for the Proposed Model.**

| Hyperparameter | Optimal Value |
| --- | --- |
| Optimizer | Adam |
| Dropout Rate | 0.5 |
| Batch Size | 128 |
| Epochs | 500 |

**Table 2. FusionNet-CXG layer architecture.**

| Layer | Parameter |
| --- | --- |
| Input | shape: (X_train_encoded.shape [1], X_train_encoded.shape [2]) |
| Batch normalization | No trainable parameters |
| Conv1D filter | filter: 64, kernel size:3, activation: ReLU. |
| Conv1D filter | filters: 128, kernel size:3, Activation: ReLU |
| XLSTM | Units: 64, dropout: 0.3, Recurrent Dropout: 0.3 |
| GRU | Units: 64, dropout: 0.3, Recurrent Dropout: 0.3 |
| dense | Units: 64, activation: ReLU, Kernel Regularizer: L2(0.01) |
| dropout | Rate: 0.5 |
| dense | Units: 128, activation: ReLU, Kernel Regularizer: L2(0.01) |
| dropout | Rate: 0.5 |
| dense | Units: 32, activation: ReLU, Kernel Regularizer: L2(0.01) |
| dropout | Rate: 0.5 |
| dense | Units: 8, activation: ReLU, Kernel Regularizer: L2(0.01) |
| dropout | Rate: 0.5 |
| dense (Output) | Units: 2, activation: Softmax (for binary classification) |

**Table 3. Proposed FusionNet-CXG model architecture and parameters.**

**FusionNet-CXG**

| Layer (type) | Output Shape | Param # |
|---|---|---|
| Batch-normalization | (none,1,20) | 80 |
| conv1D(Conv1D) | (none,1,64) | 3904 |
| conv1D_1(Conv1D) | (none,1,128) | 24704 |
| XLSTM | (none,1,64) | 49408 |
| GRU | (none,64) | 24960 |
| dense | (none, 64) | 4160 |
| dropout | (none,64) | 0 |
| dense_a1 | (none,128) | 8320 |
| dropout_a1 | (none,128) | 0 |
| dense_a2 | (none,32) | 4128 |
| dropout_a2 | (none,32) | 0 |
| dense_a3 | (none,8) | 264 |
| dropout_a3 | (none,8) | 0 |
| dense_a4 | (none,2) | 18 |

Totalparams:119937(468.50KB).

Trainable params: 119897 (468.35 KB).

Non-trainable params: 40 (160.00 KB).

## 3.8. Evaluation metrics

The model endures training for up to 500 epochs with an early stopping criterion to prevent overfitting. With each epoch, accuracy improves while model loss diminishes. Various types of statistical methods are utilised to assess the suggested model, with accuracy, F1 score, recall, specificity, sensitivity, and precision. The following are the main assessment metrics used for two class classification [51–54]. The mathematical formulae for various statistical methods used in model assessment are delineated in Eqs (15–20).

$$Precision = \frac{TP}{TP + FP} \tag{15}$$

$$Accuracy = \frac{TP + TN}{TN + FN + TP + FP} \tag{16}$$

$$Recall = \frac{TP}{FN + TP} \tag{17}$$

$$F1\ score = \frac{2(precision \times recall)}{precision + recall} \tag{18}$$

$$Specificity = \frac{TN}{TN + FP} \tag{19}$$

$$Sensitivity = \frac{TP}{TP + FN} \tag{20}$$

Here, TP is the True Positives, which correctly predicted hypothyroid cases. TN denotes True Negatives, correctly predicted non-hypothyroid cases. FP denotes False Positives; non-hypothyroid cases incorrectly predicted as hypothyroid cases. And FN denotes False Negatives; hypothyroid cases incorrectly predicted as non-hypothyroid.

Accuracy refers to identifying both hypothyroidism and non-hypothyroidism. Precision refers to all positively labelled items. Recall refers to a positive(hypothyroid) class forecast. The F1 score is the average (moderate) of recall and precision [55].

To verify our model, we performed a statistical significance test (p<0.05) comparing FusionNet-CXG to baseline models.

## 4. Results

This section describes and analyses the empirical results obtained from the FusionNet-CXG model, including a performance evaluation, a comparative analysis with established baseline models, and an in-depth discussion of the outcomes. The model underwent assessment through a multitude of performance metrics, including accuracy, F1-score, recall, specificity, and AUC-ROC, to facilitate a thorough and comprehensive evaluation.

### 4.1. Experimental analysis

Experiments were conducted in a Python 3 Google Compute Engine environment with 334.6GB of RAM and 225.3GB of disc space. This research used many machine learning libraries such as NumPy, pandas, TensorFlow, Keras, Scikit-learn, Matplotlib, Seaborn, and keras_tuner. We evaluated performance based on accuracy, F1 score, recall, and precision.

Tables 2 and 3 illustrate how the hybrid architecture is ideal for accurate hypothyroidism prediction because it uses convolutional and recurrent layers to manage the complicated sequential and non-linear nature of hypothyroid-related data. Tables 4 and 5 summarise the experimental setup for FusionNet-CXG. Table 4 covers the dataset, preprocessing, models, hyperparameters, validation, and metrics; Table 5 details the computational environment (Colab, libraries, CPU and RAM, storage).

Table 6 shows our end-to-end pipeline. We first clean the data (impute missing values), one-hot encode categorical fields, standardize numeric features, and rebalance classes with SMOTE-NC inside the training folds to avoid leakage. A stacked autoencoder reduces the inputs to a 17-dimensional embedding, which feeds the FusionNet-CXG model (Conv1D→XLSTM→GRU→Dense) trained with dropout/L2 regularisation, Adam (learning rate=0.001), batch sizes of 64–256, and up to 500 epochs with early stopping (restoring the best weights). Performance is estimated with stratified 5-fold and 5-fold stratified cross-validation repeated 10 times (50 runs), and we report standard metrics (Accuracy, Precision, Recall, F1, Specificity, ROC-AUC), examine generalization gap and best-epoch distribution, provide SHAP explanations, and check subgroup results (e.g., by sex) for fairness.

### 4.2. Effectiveness of autoencoder-based feature extraction after class balancing

To determine the consistency and robustness of feature extraction across different class balancing approaches, we trained autoencoders on each balanced training dataset. Each autoencoder was trained with early stopping (monitoring validation MSE and restoring the best weights), so the epoch count served as a maximum and training halted once the loss plateaued. Fig 7 shows the training and validation loss for autoencoders trained on datasets balanced with random over-sampling, random under-sampling, ADASYN, Borderline-SMOTE, and SMOTE-NC. In every case, the mean-squared error dropped quickly and then plateaued for both splits, consistent with effective dimensionality reduction and minimal overfitting. The results indicate that the learned features generalise across balancing approaches and are appropriate for follow-on classifiers.

**Table 4. Experimental Configuration and Pipeline Summary (Hypothyroid Binary Classification).**

| Aspect | Configuration |
|---|---|
| Dataset | Hypothyroid dataset ("hypothyroid new.csv"), binary target (P vs N classes). |
| Preprocessing | Missing values imputed (mean for numeric, mode for categorical); One-hot encoding for categorical variables; StandardScaler for normalization. |
| Balancing | SMOTE-NC applied on categorical + numeric features (where applicable). |
| Feature Extraction | Autoencoder with input = original features, latent dimension = 17 (encoded features). |
| Hybrid Models | CNN, LSTM, GRU, and FusionNet-CXG variants (Conv1D→XLSTM→GRU→Dense). |
| Regularization | Dropout (0.3–0.6), L2 penalty (0.01). |
| Optimizer | Adam, learning rate = 0.001. |
| Batch Size | 64–256 (depending on experiment). |
| Epochs | Up to 500 (with EarlyStopping on validation loss, patience = 8–20). |
| Validation Strategy | 5-fold and 10 repetitions of 5-fold stratified cross-validation |
| Evaluation Metrics | Accuracy, Precision, Recall, F1-score, Specificity (TNR), ROC–AUC, PR curve, Brier score, SHAP interpretability. |

**Table 5. Software and hardware configuration.**

| Category | Details |
|---|---|
| Programming Language | Python 3.8+ |
| Deep Learning Frameworks | TensorFlow 2.12, Keras API |
| Machine Learning & Data Processing | Scikit-learn 1.3, Imbalanced-learn (SMOTE, SMOTE-NC, ADASYN) |
| Interpretability | SHAP |
| Statistical Analysis | SciPy 1.11, NumPy, Pandas |
| Visualization Tools | Matplotlib, Seaborn |
| Execution Platform | Google Colab Environment |
| Hardware (CPU) | 2 × vCPUs (Intel Xeon, 2.20 GHz) |
| Memory (RAM) | 12.67GB |
| Operating System | Linux (Ubuntu, Google Colab default) |

Fig 8 shows the correlation heatmap between the original features (Y-axis) and the autoencoder-encoded features (X-axis). The color map ranges from blue (negative correlation) to red (positive correlation), with values clipped to ±0.2 to highlight subtle but important relationships. The latent features do not simply replicate any single original variable. Instead, each encoded feature relates to several input attributes, showing that the reduction step is capturing shared clinical structure rather than isolated signals. In practical terms, the latent space keeps the core thyroid-related information, trims repeated content, and produces compact representations that remain useful for downstream modelling, interpretation, and prediction.

### 4.3. Performance outcomes with deep learning models

The results that were obtained with the use of the deep-learning models are discussed in this section. The hyperparameter analysis of CNN+LSTM, CNN + BiLSTM and FusionNetCXG is presented in Figs 9, 11, and 13 and respectively the training and validation curves in Figs 10, 12 and 14. The models were trained with a maximum of 500 epochs using mid-stream validation loss, training could be terminated sooner when there was no longer improvement. During the early epochs the models had more loss and less accuracy yet both improved with improvement of training. CNN+LSTM and CNN + BiLSTM have

**Table 6. Experimental pipeline and methods (FusionNet-CXG).**

| Stage | Methods applied | Purpose |
|---|---|---|
| **Data Preprocessing** | • Missing values imputed (mean for numeric, mode for categorical)<br>• One-hot encoding for categorical variables<br>• StandardScaler (z-score normalization) for continuous variables<br>• SMOTE-NC applied for imbalanced classes | Ensure data quality, uniform scale, and balanced class distribution without leakage |
| **Feature Extraction** | • Stacked Autoencoder (input: original features→latent space of 17 dimensions) | Dimensionality reduction, noise removal, and representation learning |
| **Model Training** | • FusionNet-CXG: Conv1D→XLSTM→GRU→Dense – Dropout (0.3–0.6) and L2 (0.01) regularization – Adam optimizer (lr = 0.001) – Batch size = 64–256<br>• Epochs = up to 500<br>• Early stopping (patience = 8–20, restore best weights) | Train robust hybrid deep neural network while preventing overfitting |
| **Validation Strategy** | • 5-fold cross-validation<br>• 5-fold stratified cross-validation repeated 10 times (50 runs) | Reliable model evaluation with mean ± SD and 95% confidence intervals |
| **Results Post-Processing** | • Metrics: Accuracy, Precision, Recall, F1-score, Specificity, ROC–AUC<br>• Generalization gap (train vs val acc/loss)<br>• Best epoch distribution (early stopping)<br>• SHAP-based interpretability | Comprehensive performance reporting, and model interpretability |
| **Subgroup Analysis** | • Performance evaluation across demographic subgroups (e.g., sex attribute) | Assess fairness and demographic-specific generalisation |

0.9192 and 0.9203 mean accuracies respectively under the condition of 5-fold stratified cross-validation repeated 10 times, and FusionNet-CXG reached the highest mean accuracy of 0.9394. The performance of the three models is compared on the same repeated cross-validation scenario and Fig 16 indicates that FusionNet-CXG provided the best overall result.

## 4.4. Confusion matrix and scatter plot result analysis

The confusion matrix study evaluates the strengths and shortcomings of deep learning algorithms for diagnosis. Performance evaluation using the confusion matrix is depicted in Fig 6. Meanwhile, Fig 16 presents accuracy comparisons of deep learning models.

## 4.5. Experimental results of deep neural networks

Fig 9 shows how the CNN-LSTM model behaves under different hyperparameter settings. Each point gives the test accuracy for one setting, and the shaded region shows the corresponding 95% Wilson confidence interval. The x-axis labels summarize the setting used in each run, where A stands for Adam and R stands for RMSprop, followed by dropout rate, batch size, and number of epochs.

Across the tested settings, the model accuracy stays in a narrow range, mostly around 91% to 92%. This means the model performance is fairly stable and does not change sharply when the hyperparameters are varied within the tested range. The highest accuracy appears for the RMSprop setting with dropout 0.5, batch size 32, and 100 epochs, while the lowest value is seen for the Adam setting with dropout 0.3, batch size 64, and 100 epochs. Overall, the figure suggests that CNN-LSTM gives consistently strong performance across different configurations, with only small differences between settings.

Fig 10(a) shows a comparison of the training and validation loss curves of the CNN+LSTM model. Each run was allowed to go for a maximum of 500 epochs, and early stopping was used according to validation loss and the best weights were restored. The training loss was large in the first stage, however, it reduced quickly, and the validation loss showed a similar tendency. After the early phase, both curves came close to each other and they became almost stable, which was an indication of convergence of the model. Fig 10(b) shows the corresponding accuracy of the training and

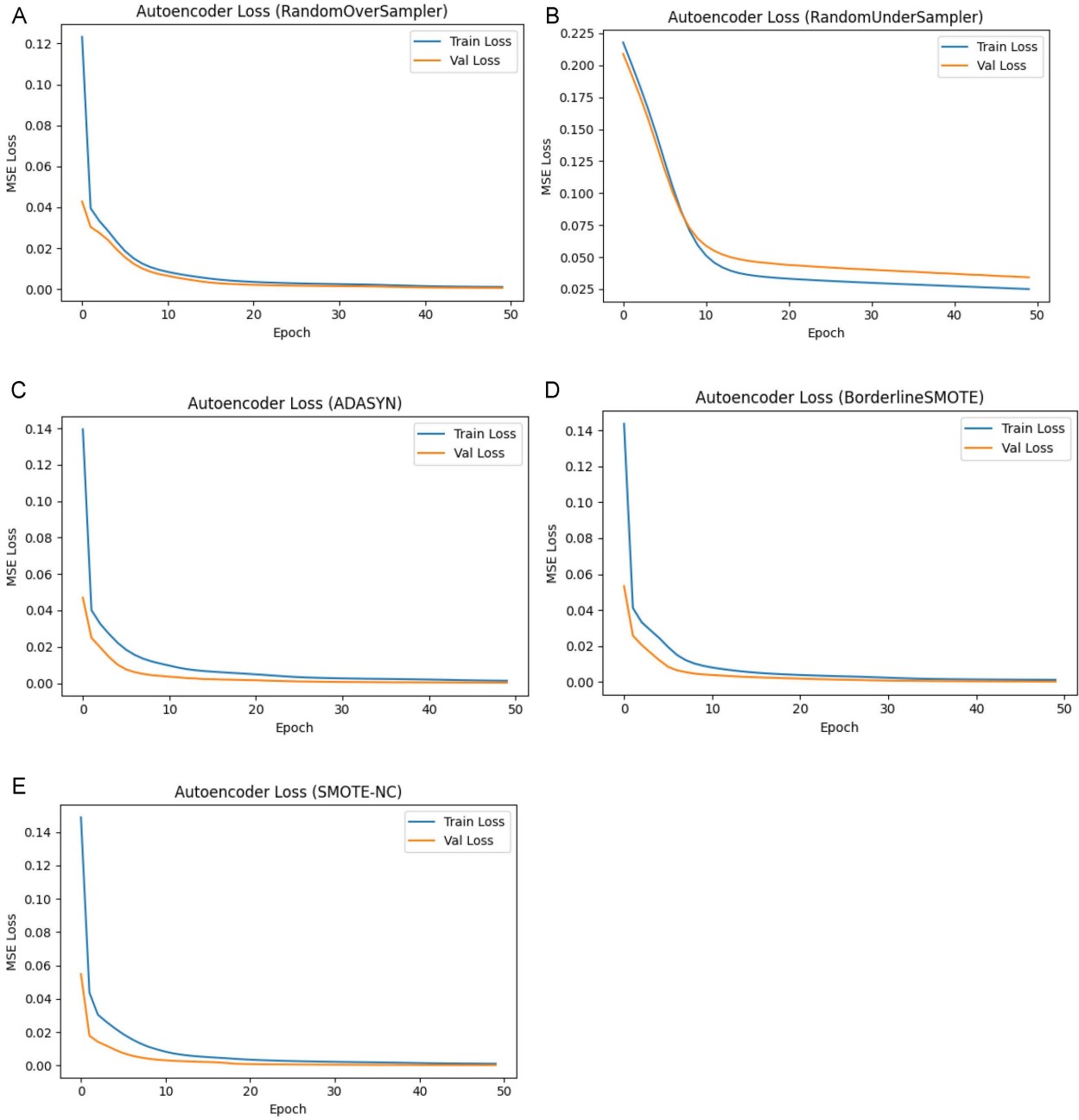

**Fig 7. Autoencoder training and validation loss curves for different class balancing approaches: (a) Random Oversampling (b) Random Under sampling (c) ADASYN (d) Borderline-SMOTE (e) SMOTE-NC.**

validation sets per epoch. The training accuracy steadily increased, and then it reached a steady level, and the validation accuracy only experienced slight fluctuations around 90%–92%. The maximum validation accuracy achieved during training was 91.92%, which represents the best validation accuracy that can be achieved by the model.

Fig 11 shows the test accuracy for the different hyperparameter settings. The x-axis lists the tested combinations, and the blue line marks the accuracy obtained for each one. The shaded area and error bars show the 95% Wilson confidence

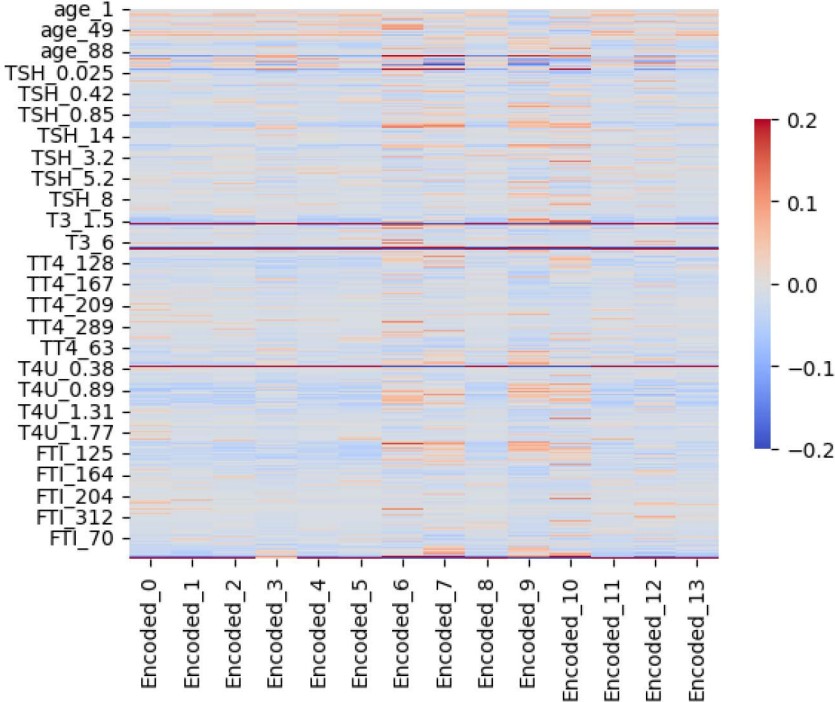

**Fig 8. Correlation between original features and encoded features.**

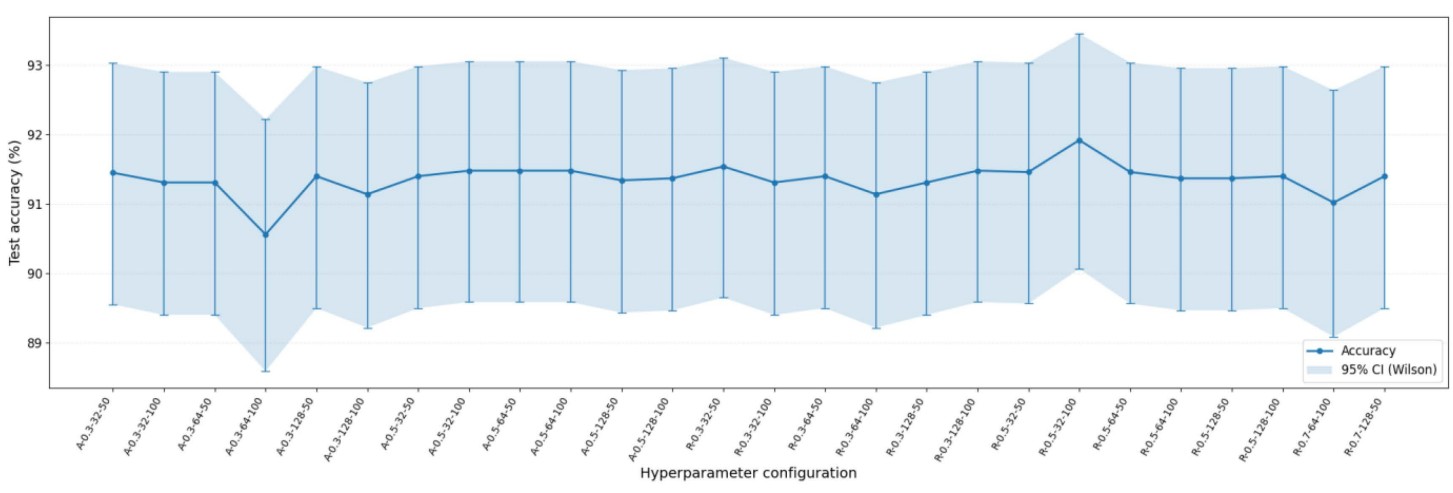

A = Adam, R = RMSprop; format = Optimizer-Dropout-Batch size-Epochs

**Fig 9. Illustrates the test Accuracy across hyperparameter tuning results of CNN-LSTM.**

intervals. In the earlier settings, the accuracy stays mostly in the high-80s to around 90%. In the later settings, several configurations give higher values in the 91%–92% range. The best result is 92.03%. Overall, the figure shows that performance changes with the selected hyperparameters, but some settings give clearly better accuracy than others.

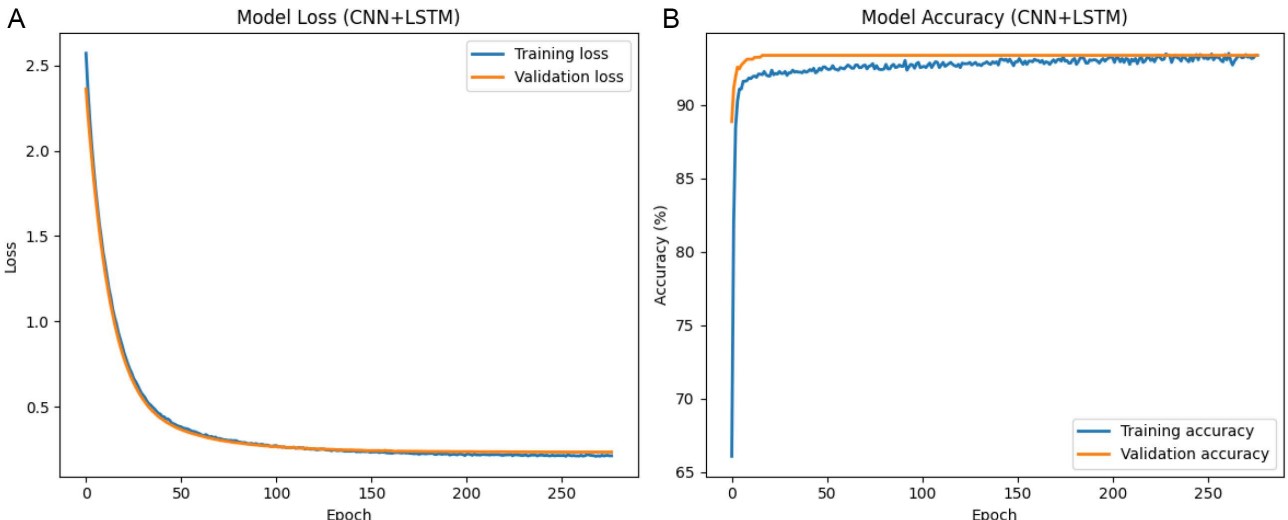

**Fig 10. (a) Training and validation loss of CNN-LSTM classifier over epochs. (b) accuracy score between training and validating data over epochs of CNN-LSTM classifier.**

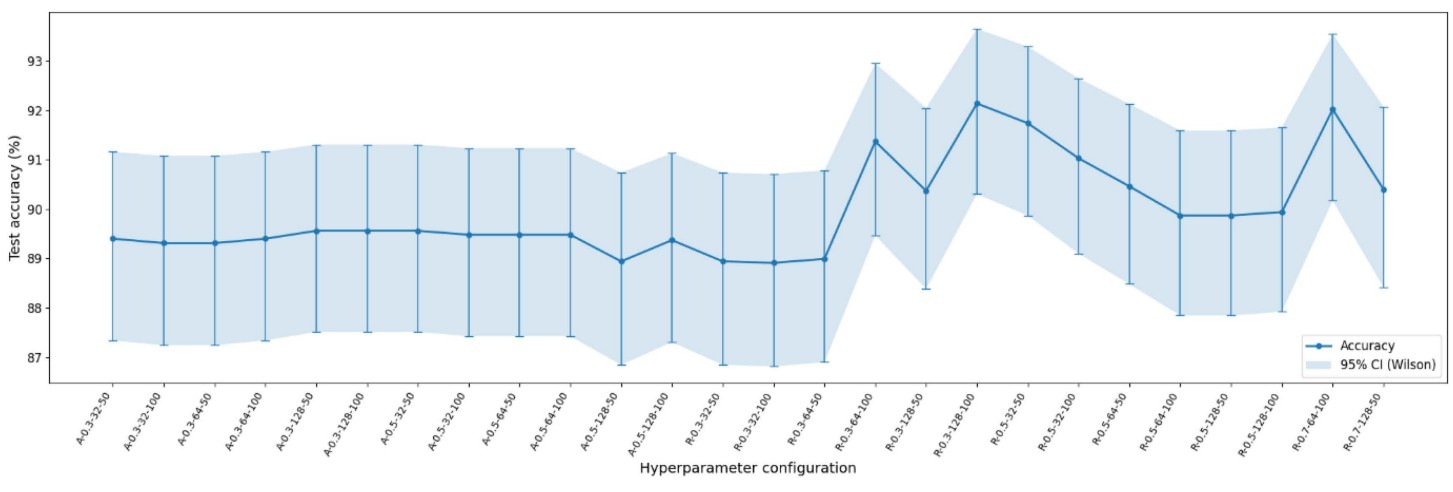

A = Adam; R = RMSprop; label format = Optimizer-Dropout-Batch size-Epochs

**Fig 11. Depicts the test Accuracy across Hyperparameter Configurations for CNN+BiLSTM.**

The CNN+BiLSTM model learning dynamics are shown in Fig 12. Fig 12(a) indicates that the training and validation loss decrease drastically at the beginning of the process and then level off, and they stay close to each other during the training process. Such behaviour signifies acceptable convergence. The training and validation accuracy in Fig 12(b) improve at very high rates in the first epochs and then change by relatively small values. The accuracy of validation is like that of the training accuracy, with a value of about 92.03, which shows that there is a stable generalisation in training.

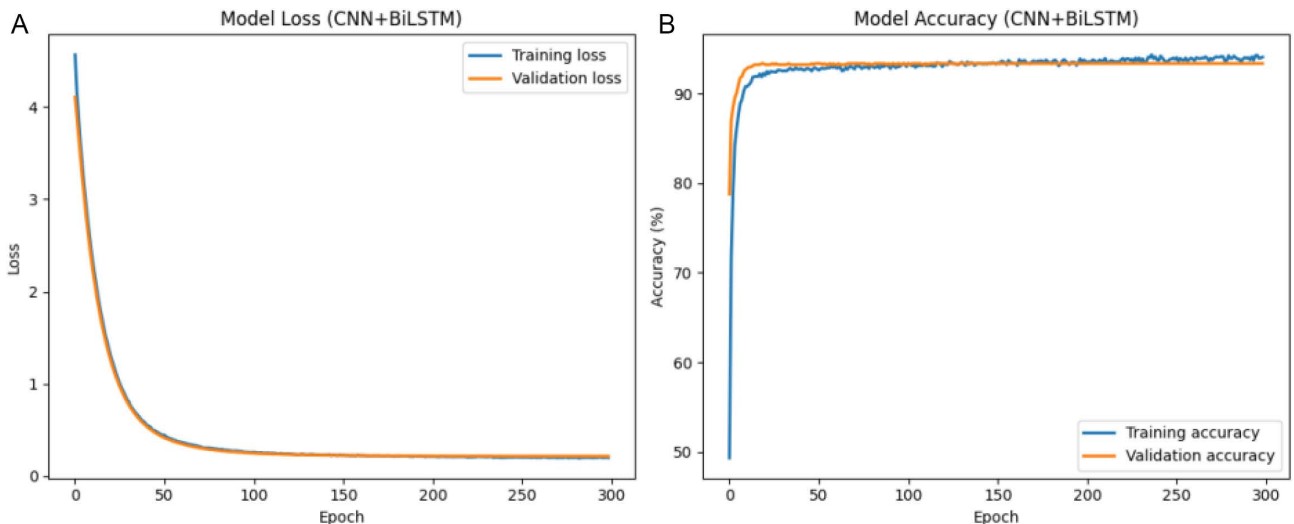

**Fig 12. (a) Training and validation loss of the CNN+BiLSTM classifier. Fig12 (b) training and validation accuracy over epochs of the CNN+BiLSTM classifier.**

## 4.6. Experimental outcomes of proposed FusionNet-CXG model

Following empirical validation, it was discovered that the proposed model was the best predictor for the hypothyroid illness identification dataset. However, it was felt that more superior and trustworthy outcomes were necessary.

Fig 13 shows the test accuracy of FusionNet-CXG under different hyperparameter settings, together with 95% Wilson confidence intervals. The x-axis labels summarize the optimizer, dropout rate, batch size, and number of epochs used in each run. The accuracy values stay within a narrow range, mostly around 92% to 93%. This shows that the model performs consistently across the tested settings. The differences between neighbouring configurations are small, and many

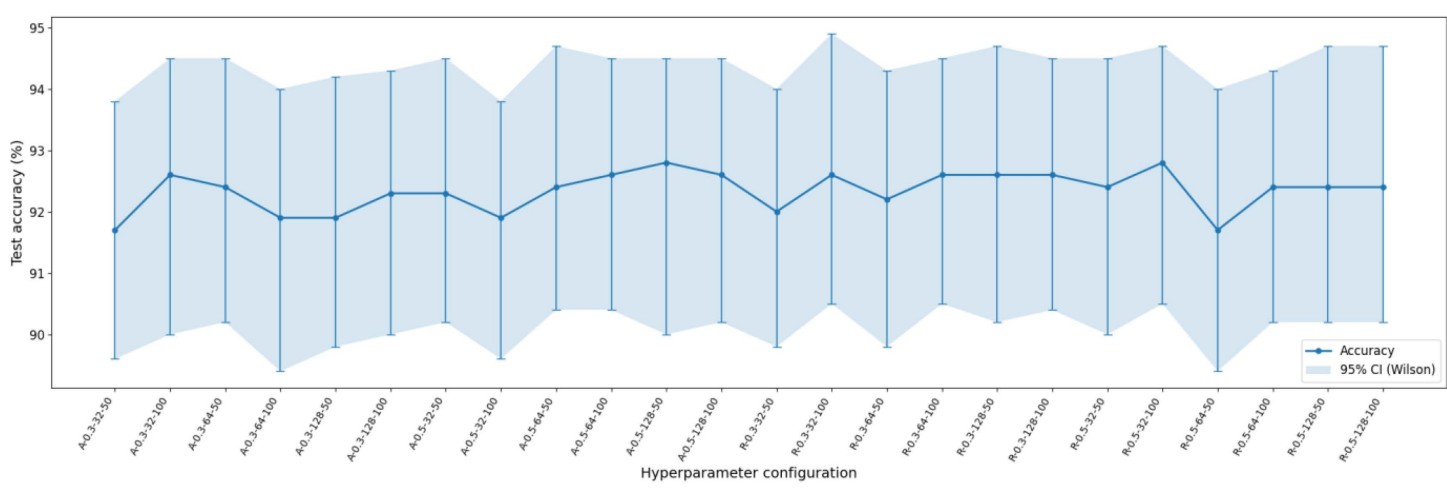

A = Adam; R = RMSprop; label format = Optimizer-Dropout-Batch size-Epochs

**Fig 13. Test accuracy across different hyperparameter configurations for the FusionNet-CXG model.**

of the confidence intervals overlap. Overall, the figure suggests that FusionNet-CXG remains stable across the evaluated hyperparameter combinations.

Fig 14(a) displays the training and validation loss for the FusionNet-CXG. The model was trained for up to 500 epochs and employed early stopping based on validation loss. The training loss is large at the beginning, but it drops very rapidly and in a very similar way to the validation loss. After this initial steep drop, the losses only fluctuate slightly, indicating that the model has converged. Fig 14(b) depicts the training and validation accuracy. They shoot up rapidly at the beginning and then remain almost constant in the latter epochs. The validation accuracy reaches 92.83% and remains close to the training accuracy. Thus, the model learns the data rapidly, and no overfitting is observed.

Fig 15 compares model accuracy between female and male groups using 95% Wilson confidence intervals. The accuracy was 0.9122 for females (95% CI: 0.8841–0.9339) and 0.9446 for males (95% CI: 0.9210–0.9615), while the overall accuracy was 0.9284. The dashed line indicates the overall accuracy, and the semi-transparent shaded band represents its 95% confidence interval. Although the male group showed slightly higher accuracy, the confidence intervals were close, suggesting that the model performed similarly across both groups.

## 4.7. FusionNet-CXG performance (5-Fold Vs. 5-fold stratified cross-validation repeated 10 times)

In 5-fold CV, FusionNet-CXG attained 0.9284 accuracy, 0.9284 precision, 1.0000 recall, and 0.9601 F1, providing a strong baseline. In the more rigorous 5-fold stratified cross-validation repeated 10 times (50 runs), performance remained high and consistent: accuracy = 0.9394 ± 0.0342 (95% CI 0.9299–0.9489), F1 = 0.9066 ± 0.1120 (95% CI 0.8755–0.9377), precision = 0.9321 ± 0.0193 (95% CI 0.9268–0.9375), and recall = 0.9089 ± 0.1878 (95% CI 0.8568–0.9610). Taken together, these results demonstrate robust generalisation across resamples, with consistently high precision and recall and a correspondingly strong F1.

Table 7 provides a full descriptive summary of FusionNet-CXG over 50 runs, including mean ± SD, 95% confidence intervals, and distributional descriptors (min–max, median, IQR). Accuracy is concentrated near the upper range (mean 0.9394, median 0.9525, IQR 0.0076), with a few low outliers (min 0.7662). Precision is high and very stable (mean 0.9321, SD 0.0193, IQR 0.0001). The model shows high accuracy and stable precision, while recall reaches its maximum in most

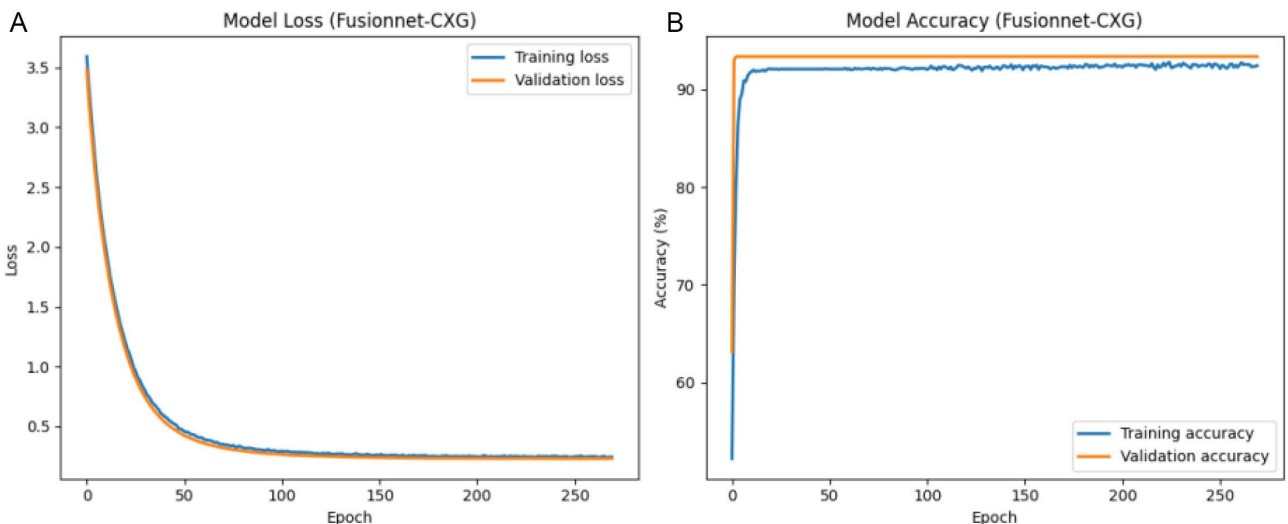

**Fig 14. (a) Training and validation loss of the FusionNet-CXG model. Fig 14 (b) validation and training accuracy of the FusionNet-CXG model.**

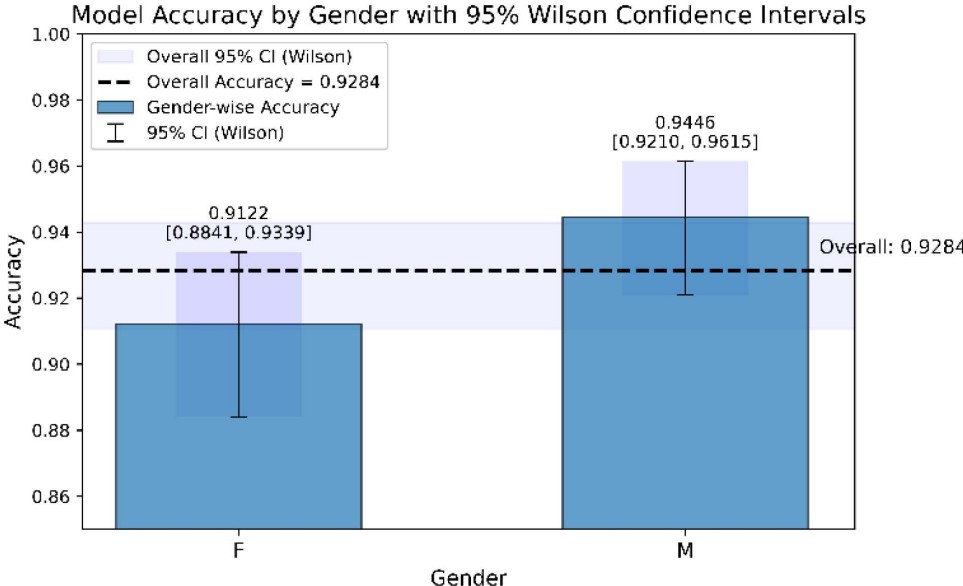

**Fig 15. Model accuracy among different genders (with 95% Wilson CI).**

folds (median 1.0), underscoring reliable detection. Rare dips increase F1 variability (mean 0.9066), but the central tendency remains strong. Table 8 condenses the 50-fold evidence into mean ± SD and 95% CIs.

## 4.8. Comparative performance of deep learning models

This section examines the evaluation metrics of FusionNet-CXG, a novel deep-learning technique. Fig 16 compares the mean performance of the deep learning models based on cross-validation test-fold results under 5-fold stratified cross-validation repeated 10 times, in terms of accuracy, precision, recall, and F1-score. FusionNet-CXG shows the strongest overall performance among the compared models.

Table 9 compares the performance of CNN+LSTM, CNN+BiLSTM, and the proposed approach for hypothyroid classification. Among the three models, the proposed approach achieved the strongest overall performance, with the highest accuracy (0.9394 ± 0.0342; 95% CI: [0.9299, 0.9489]), AUC-ROC (0.9400 ± 0.0310; 95% CI: [0.9314, 0.9486]), sensitivity (0.9200 ± 0.0450; 95% CI: [0.9075, 0.9325]), and specificity (0.9730 ± 0.0280; 95% CI: [0.9652, 0.9808]). CNN+BiLSTM showed a slight but consistent improvement over CNN+LSTM across all reported metrics, indicating a modest benefit of bidirectional temporal modeling over the conventional LSTM structure. However, both baseline models remained inferior to the proposed approach. The favorable confidence intervals of the proposed model further suggest that its performance was not only better on average but also stable across repeated evaluations. Overall, these results indicate that the proposed approach provides a more reliable and effective framework for hypothyroid classification than the baseline deep learning models.

Table 10 presents the statistical comparison of mean classification accuracies between the proposed FusionNet-CXG model and the baseline models using a paired t-test. Within the 10 repetitions of the 5-fold stratified cross-validation framework, the accuracies obtained from the five folds in each repetition were first averaged to generate one mean accuracy per repetition for each model. These 10 repetition-wise mean accuracies were then used for statistical testing, thereby avoiding the treatment of individual fold-wise results as statistically independent observations.

**Table 7. Overall performance of FusionNet-CXG across 50 runs (mean±SD; 95% CI).**

| Metrics | Mean | SD | 95% CI Low | 95% CI High | Min | Max | Median | IQR |
|---|---|---|---|---|---|---|---|---|
| Accuracy | 0.9394 | 0.0342 | 0.9299 | 0.9489 | 0.7662 | 0.9652 | 0.9525 | 0.0076 |
| F1 | 0.9066 | 0.112 | 0.8755 | 0.9377 | 0.5703 | 0.9601 | 0.96 | 0.0007 |
| Precision | 0.9321 | 0.0193 | 0.9268 | 0.9375 | 0.9219 | 0.9812 | 0.9231 | 0.0001 |
| Recall | 0.9089 | 0.1878 | 0.8568 | 0.961 | 0.4052 | 1 | 1 | 0 |

**Table 8. Overall Performance of FusionNet-CXG (Mean±SD; 95% CI).**

| Metric | Mean±SD | 95% CI |
|---|---|---|
| Accuracy | 0.9394±0.0342 | [0.9299, 0.9489] |
| F1 | 0.9066±0.1120 | [0.8755, 0.9377] |
| Precision | 0.9321±0.0193 | [0.9268, 0.9375] |
| Recall | 0.9089±0.1878 | [0.8568, 0.9610] |

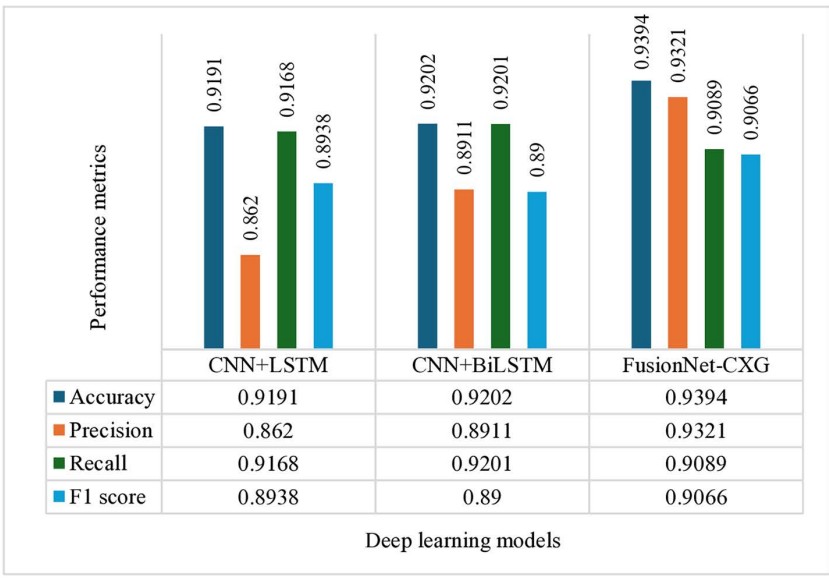

| | CNN+LSTM | CNN+BiLSTM | FusionNet-CXG |
|---|---|---|---|
| ■ Accuracy | 0.9191 | 0.9202 | 0.9394 |
| ■ Precision | 0.862 | 0.8911 | 0.9321 |
| ■ Recall | 0.9168 | 0.9201 | 0.9089 |
| ■ F1 score | 0.8938 | 0.89 | 0.9066 |

Deep learning models

**Fig 16. Mean performance comparison of the deep learning models on cross-validation test folds under 5-fold stratified cross-validation repeated 10 times.**

Before applying the paired t-test, the normality of the paired differences was assessed using the Shapiro–Wilk test, as summarized in Table 11. The results showed no significant deviation from normality for either comparison (p>0.05), indicating that the normality assumption was satisfied. Because the paired t-test is based on within-pair differences, homogeneity of variances between the compared models was not a required assumption in this analysis.

In addition, the Wilcoxon signed-rank test was performed as a non-parametric robustness check. The Wilcoxon results were consistent with those of the paired t-test, further supporting the statistical significance of the findings.

FusionNet-CXG achieved a higher mean accuracy than CNN+LSTM (0.9394 vs. 0.9191) and CNN+BiLSTM (0.9394 vs. 0.9202). The corresponding paired t-test p-values of 0.031 and 0.018 indicate that these improvements were

**Table 9. Comparative Performance of CNN+LSTM, CNN+BiLSTM, and Proposed Model with Mean±SD and 95% Confidence Intervals.**

| Model | Metric | Mean±SD | 95% CI |
|---|---|---|---|
| CNN+LSTM | Accuracy | 0.9191±0.0340 | [0.9097, 0.9285] |
| | AUC-ROC | 0.9102±0.0320 | [0.9013, 0.9191] |
| | Sensitivity | 0.8705±0.0400 | [0.8594, 0.8816] |
| | Specificity | 0.9003±0.0350 | [0.8906, 0.9100] |
| CNN+BiLSTM | Accuracy | 0.9202±0.0320 | [0.9113, 0.9291] |
| | AUC-ROC | 0.9204±0.0300 | [0.9121, 0.9287] |
| | Sensitivity | 0.8906±0.0380 | [0.8801, 0.9011] |
| | Specificity | 0.9104±0.0340 | [0.9010, 0.9198] |
| Proposed Approach | Accuracy | 0.9394±0.0342 | [0.9299, 0.9489] |
| | AUC-ROC | 0.9400±0.0310 | [0.9314, 0.9486] |
| | Sensitivity | 0.9200±0.0450 | [0.9075, 0.9325] |
| | Specificity | 0.9730±0.0280 | [0.9652, 0.9808] |

**Table 10. Statistical comparison based on 10 repetition-wise mean accuracies obtained from 10×5 repeated stratified cross-validation.**

| Comparison | Mean Accuracy (FusionNet-CXG) | Mean Accuracy (Baseline) | Paired t-test p-value | Interpretation |
|---|---|---|---|---|
| FusionNet-CXG vs CNN+LSTM | 0.9394 | 0.9191 | 0.031 | Significant improvement with FusionNet-CXG |
| FusionNet-CXG vs CNN+BiLSTM | 0.9394 | 0.9202 | 0.018 | Significant improvement with FusionNet-CXG |

**Table 11. Verification of Assumptions for Paired t-Test.**

| Comparison | Data Used | Normality Test (Shapiro–Wilk) | Normality Satisfied | Additional Test (Wilcoxon) | Conclusion |
|---|---|---|---|---|---|
| FusionNet-CXG vs CNN+LSTM | 10 repetition means | 0.112 | Yes | 0.028 | Results are consistent and significant |
| FusionNet-CXG vs CNN+BiLSTM | 10 repetition means | 0.089 | Yes | 0.021 | Results are consistent and significant |

statistically significant at the 5% level. These results suggest that the observed performance gains are unlikely to be attributable to random variation alone and support the improved performance of the proposed model.

## 4.9. XAI results analysis

Fig 17 shows the results of the XAI analysis using the proposed method. This analysis establishes the significance of each dataset feature in the decision-making process of the proposed method for hypothyroid diagnosis. The SHAP chart analysis shows the importance scores of dataset features in order from highest to lowest. This analysis shows that the proposed model uses T3, FTI, TSH, T3_measured, T4U, age, TBG, and on_thyroxine to help diagnose hypothyroid disease.

SHAP indicated that T3, T4U, TSH, and FTI had the strongest influence on hypothyroidism predictions, aligning the model's outputs with clinically expected markers. T3 (triiodothyronine) is the hormonally active form, whereas TT4 (total thyroxine) reflects the total circulating thyroxine level. Changes in T3 and TT4 are strongly associated with thyroid dysfunction, since abnormal levels can affect metabolic activity, heart rate, thermoregulation, and perceived energy. Among these markers, T3 is often especially informative because it acts at the tissue level and directly influences how cells use and regulate energy. Elevated T3 levels can mean hyperthyroidism, and reduced T3 levels can mean hypothyroidism.

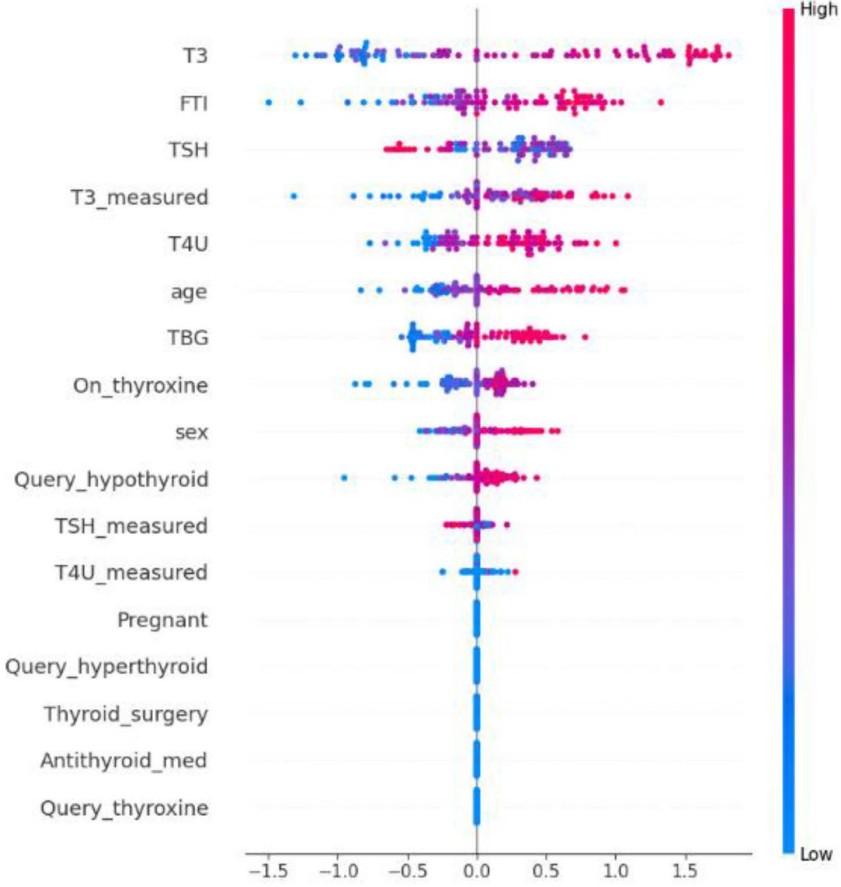

**Fig 17. SHAP-Based Explainability of the Proposed Hypothyroid Diagnostic Model.**

This makes T3 a very important test for telling the difference between thyroid conditions. TT4, because it captures both protein-bound and unbound thyroxine, offers a broad indicator of how well the thyroid is functioning, helping to detect problems and monitor overall thyroid health. When considered alongside TSH (the pituitary signal that regulates thyroid activity) and the Free Thyroxine Index (FTI), these markers together provide a comprehensive view of thyroid status. Grounding predictions in standard thyroid markers makes the model transparent to clinicians and increases its utility at the point of care.

### 4.10. Comparison with state-of-the-art studies

Table 12 compares a wide range of recent thyroid-classification methods, including machine-learning, deep-learning, and optimisation-driven models, together with their reported accuracy and precision. Among optimisation approaches, the hybrid DE-BOA+FCM framework showed the strongest results, reaching 94.3% accuracy and 89.78% precision [56], which is higher than the individual DE, Butterfly Optimisation, and FCM models. Deep-learning methods also continue to show competitive performance; for instance, multi-channel CNN variants reported by Zhang et al. [57] achieved 90.2% accuracy, while other CNN architectures generally performed better than traditional SVM classifiers [58–60]. The ATSO-DeepCNN-GWO model further raised the accuracy to 92% with a precision of 94% [6]. Recurrent neural networks such as LSTM, BiLSTM, GRU, and ELM usually produced moderate accuracy values in the 86–88% range [23]. Traditional

machine-learning models such as Random Forest have also been explored, although one study reported a comparatively lower accuracy of 82.4% [3]. Ensemble-based feature-selection pipelines, including those using Random Forest, PCA, and RFE, achieved 92.2% accuracy and 90.3% precision [62], and recent boosting-based methods, such as Gradient Boosting Machine, XGBoost, and Random Forest, also showed competitive performance on alternate datasets [63].

To enable a fair comparison in this study, the baseline CNN+LSTM and CNN+BiLSTM models were re-implemented and trained on the same hypothyroid dataset, using identical preprocessing, class-balancing, feature-extraction, and validation protocols. Several works summarized in Table 12, such as [23] and [62], rely on different data modalities (e.g., imaging or genomic datasets); these studies are included to illustrate methodological diversity in the literature rather than to suggest strict numerical comparability. When all models were assessed under the same experimental conditions, the proposed FusionNet-CXG achieved the highest and most consistent performance, with an accuracy of 93.94% and a precision of 93.21%. For clarity, a note has been added below Table 12 specifying which studies used datasets that differ from the hypothyroid data analyzed in this work.

## 5. Discussion

### 5.1. Hyperparameter tuning and optimization

Hyperparameter tuning was critical for FusionNet-CXG. GridSearchCV was employed to explore combinations of optimizer, dropout rate, batch size, and maximum number of training epochs. The best-performing configuration used Adam, dropout = 0.5, batch size = 128, and a 500-epoch cap with early stopping (training typically stopped much earlier). Fig 13 shows accuracy trends across hyperparameter settings, with dropout = 0.5 and batch size = 128 yielding the most consistent learning.

**Table 12. Performance comparison of the Proposed model with state-of-the-art models.**

| Ref | Year | Models | Accuracy | Precision |
|---|---|---|---|---|
| [56] | 2022 | Differential Evolution (DE) <br> Butterfly Optimization <br> Fuzzy C-Means <br> DE-BOA+FCM | 88.4% <br> 90.6% <br> 89.9% <br> 94.3% | 58.45% <br> 78.12% <br> 82.45% <br> 89.78% |
| [57] | 2022 | CNN | 90.2% | 94.4% |
| [58] | 2023 | CNN | 89% | 87% |
| [59] | 2023 | SVM | 86% | 84% |
| [60] | 2023 | SVM | 84.72% | – |
| [6] | 2023 | ATSO-DeepCNN-GWO | 92% | 94% |
| [3] | 2023 | RF | 82.4%. | – |
| [23] † | 2024 | LSTM <br> BiLSTM <br> GRU <br> ELM | 86% <br> 88% <br> 85% <br> 86% | Sensitivity-83%, <br> 85% <br> 85% <br> 85% |
| [61] | 2024 | Ensemble learning mode | 90.4% | – |
| [60] † | 2025 | Random Forest, PCA, RFE | 92.2% | 90.3% |
| [63] | 2025 | Gradient Boosting Machine <br> XGBoost <br> RF | 90.91% <br> 86.36% <br> 90.91% | 92.31% <br> 91.67% <br> 92.31% |
| **This study** | **2025** | **FusionNet-CXG** | **93.94%** | **93.21%** |

Note: Entries marked with (†) correspond to studies that used datasets different from the UCI/Kaggle hypothyroid dataset employed in this work.

## 5.2. Error analysis and model robustness

A detailed error analysis was conducted to better characterise the misclassifications produced by the model. Most errors arose in borderline cases, where TSH and T3 values lay close to the hypothyroid decision threshold, making categorisation inherently difficult. This pattern suggests that incorporating additional clinical information, such as thyroid ultrasound imaging or genetic markers, could further strengthen model performance.

As shown in Fig 15, the FusionNet-CXG model also demonstrated good generalisation across gender. The classification accuracy for male patients was 94.46%, while for female patients it was 91.22%, indicating only a modest difference that may reflect hormonal variations influencing thyroid function.

## 5.3. Discussion of key findings

The experimental results highlight a few significant insights:

- Hybrid models outperform classic deep learning techniques. FusionNet-CXG, which combines CNN, XLSTM, and GRU, delivers greater accuracy and feature extraction than CNN+LSTM and CNN+BiLSTM.

- Feature selection using autoencoders increases performance by decreasing noise and picking the most relevant features, hence improving generalisation and preventing overfitting.

- SMOTE-NC efficiently reduces class imbalance – SMOTE-NC ensures a fair representation of hypothyroid patients, resulting in better memory and sensitivity.

- Hyperparameter adjustment is crucial for optimisation; proper tuning of dropout rates, learning rates, and batch sizes has a major influence on the model's stability and accuracy.

- The training and validation curves sit close to each other under early stopping (cap 500 epochs; patience = 10; best weights restored), so overfitting appears minimal across the 5-fold stratified cross-validation repeated 10 runs.

- SHAP highlights T3, TT4, TSH, FTI, and T4U as dominant contributors, with directions consistent with clinical thyroid assessment, supporting trustworthy, clinically aligned predictions.

This study on detecting hypothyroid illness has several practical constraints. First, the hypothyroid dataset originates from 1987, and its feature distribution may not fully reflect current clinical practice or contemporary screening patterns. Second, limited demographic descriptors restrict subgroup-level analysis, and the absence of temporal measurements prevents modelling disease progression over time. These factors can affect how well the reported performance transfers to broader or newer patient cohorts. Future work will focus on validating the model on more recent, multi-centre cohorts and extending the framework to longitudinal or time-series data to capture progression and improve clinical robustness. In addition, the cross-study comparison in Table 12 is intended as a contextual literature summary; because prior studies often use different datasets and evaluation protocols and frequently do not report fold-wise or repeated-run results, formal statistical significance testing across studies is not feasible from published summaries alone. Accordingly, Table 12 should be interpreted as a descriptive comparison, while uncertainty for our model is reported under a consistent experimental setup (mean±SD and confidence intervals).

## 6. Conclusion

This study proposes FusionNet-CXG, an integrated deep-learning system that incorporates CNN, XLSTM, and GRU layers to enhance hypothyroidism prediction. Using SMOTE-NC to address class imbalance and autoencoder-based feature extraction, the model achieved strong diagnostic performance, 93.94% accuracy and 0.94 AUC-ROC, under 5-fold stratified cross-validation repeated 10 times (50 runs), outperforming existing models by capturing spatial and sequential

patterns in medical data. In addition, SHAP was used to interpret feature contributions, enhancing transparency and clinician trust. However, dependence on an older dataset and limited demographics restricts generalisability, highlighting the need for validation on diverse, real-world datasets. Future work may incorporate attention processes or transfer learning; addressing these elements could further improve diagnostic accuracy and clinical outcomes.

## Author contributions

**Conceptualization:** Divya Kesavulu.

**Data curation:** Divya Kesavulu.

**Formal analysis:** Divya Kesavulu.

**Investigation:** Divya Kesavulu.

**Methodology:** Divya Kesavulu.

**Project administration:** Divya Kesavulu.

**Resources:** Divya Kesavulu.

**Software:** Divya Kesavulu.

**Supervision:** Kannadasan R.

**Validation:** Divya Kesavulu.

**Visualization:** Divya Kesavulu.

**Writing – original draft:** Divya Kesavulu.

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
