## [Decision Letter · Decision Letter 0]

26 Aug 2025

PONE-D-25-37245A Hybrid CNN-XLSTM-GRU Deep Learning Model with Autoencoder-Based Feature Selection for Hypothyroidism DiagnosisPLOS ONE

Dear Dr. R,

Thank you for submitting your manuscript to PLOS ONE. After careful consideration, we feel that it has merit but does not fully meet PLOS ONE’s publication criteria as it currently stands. Therefore, we invite you to submit a revised version of the manuscript that addresses the points raised during the review process.

In particular:

Experimental setup (including data preprocesisng and results postprocessing) should be described clearly and with more details,Analysis of measurement errors and statistical properties of the results should be included,Language problems should be removed,Figures should have all axes described precisely (also with values - see problems with Fig. 2),Figures should be clearly readable (see problems with Fig. 6, Fig. 8, Fig. 10, Fig. 12, Fig. 13),Equations should be presented and described properly (see: (9)(11), what are b_h, g_z?),Format of references should be uniform (see. [18], [23]),Authors should consider more related works which are not from India and China,Discrepancies between numbers of training epochs reported in the text, in tables and in figures should be clarified,Comparison with related results from other Authors would greatly improve the value of the manuscript.

Remark 1: Authors should not be forced to cite publications of Reviewers. Please be informed that, in such cases, Authors may cite some of the proposed publications, but only if they are relevant to the subject analyzed. Citing or not citing these publications will not influence the Editor’s decision. Reviews are presented in their original form for transparency of the review process.

Remark 2: One of the Reviewers wrote: "Authors should delete the Study Limitations section." Authors should be informed that discussion of study limitations is an important part of the manuscript and often is presented as a separated section/subsection. Authors are not encouraged to remove such part of their work.

We look forward to receiving your revised manuscript.

Kind regards,

Maciej Huk, Ph.D.

Academic Editor

PLOS ONE

3. Thank you for uploading your study's underlying data set. Unfortunately, the repository you have noted in your Data Availability statement does not qualify as an acceptable data repository according to PLOS's standards.

At this time, please upload the minimal data set necessary to replicate your study's findings to a stable, public repository (such as figshare or Dryad) and provide us with the relevant URLs, DOIs, or accession numbers that may be used to access these data. For a list of recommended repositories and additional information on PLOS standards for data deposition, please see https://journals.plos.org/plosone/s/recommended-repositories....

Additional Editor Comments (if provided):

Reviewers' comments:

Reviewer's Responses to Questions

**Comments to the Author**

1. Is the manuscript technically sound, and do the data support the conclusions?

Reviewer #1: Yes

Reviewer #2: Yes

Reviewer #3: Yes

Reviewer #4: Yes

2. Has the statistical analysis been performed appropriately and rigorously? 

Reviewer #1: Yes

Reviewer #2: Yes

Reviewer #3: Yes

Reviewer #4: N/A

3. Have the authors made all data underlying the findings in their manuscript fully available?

The PLOS Data policy requires authors to make all data underlying the findings described in their manuscript fully available without restriction, with rare exception (please refer to the Data Availability Statement in the manuscript PDF file). The data should be provided as part of the manuscript or its supporting information, or deposited to a public repository. For example, in addition to summary statistics, the data points behind means, medians and variance measures should be available. If there are restrictions on publicly sharing data—e.g. participant privacy or use of data from a third party—those must be specified.requires authors to make all data underlying the findings described in their manuscript fully available without restriction, with rare exception (please refer to the Data Availability Statement in the manuscript PDF file). The data should be provided as part of the manuscript or its supporting information, or deposited to a public repository. For example, in addition to summary statistics, the data points behind means, medians and variance measures should be available. If there are restrictions on publicly sharing data—e.g. participant privacy or use of data from a third party—those must be specified.requires authors to make all data underlying the findings described in their manuscript fully available without restriction, with rare exception (please refer to the Data Availability Statement in the manuscript PDF file). The data should be provided as part of the manuscript or its supporting information, or deposited to a public repository. For example, in addition to summary statistics, the data points behind means, medians and variance measures should be available. If there are restrictions on publicly sharing data—e.g. participant privacy or use of data from a third party—those must be specified.requires authors to make all data underlying the findings described in their manuscript fully available without restriction, with rare exception (please refer to the Data Availability Statement in the manuscript PDF file). The data should be provided as part of the manuscript or its supporting information, or deposited to a public repository. For example, in addition to summary statistics, the data points behind means, medians and variance measures should be available. If there are restrictions on publicly sharing data—e.g. participant privacy or use of data from a third party—those must be specified.

Reviewer #1: Yes

Reviewer #2: Yes

Reviewer #3: Yes

Reviewer #4: Yes

4. Is the manuscript presented in an intelligible fashion and written in standard English?

Reviewer #1: Yes

Reviewer #2: Yes

Reviewer #3: Yes

Reviewer #4: Yes

5. Review Comments to the Author

Reviewer #1: 1 The model achieves high accuracy, there is limited discussion of interpretability tools to help clinicians understand decision-making.

2. Despite the use of autoencoders, the paper would benefit from more rigorous reporting on overfitting

3. The work reliability is poor as it havent perfomed in repeated cross-validation testing.

Overall : The work is promising and methodologically solid, but its imited interpretability analysis reduces clinical applicability also adding interpretability/robustness analyses would significantly improve the manuscript’s impact.

Reviewer #2: 1. The motivations of the paper are not clear.

2. Contributions are not mentioned. Most importantly, the structure of the Introduction section is very poor.

3. Related schemes are not discussed properly. The following references may be cited:

A Smart Healthcare System Using Consumer Electronics and Federated Learning to Automatically Diagnose Diabetic Foot Ulcers

Drug Target Interaction Prediction Using Machine Learning Techniques – A Review

HCNNet: hybrid convolution neural network for automatic identification of ischaemia in diabetic foot ulcer wounds

Analysis and Prediction of COVID-19 Using Growth Analysis Models: A Case Study

Role of internet of things and machine learning in smart healthcare

Early prediction of cognitive impairments using physiological signal for enhanced socioeconomic status

4. The proposed scheme is unstructured. It is hard to identify the novelty of the proposed work.

5. Equations and figures are not represented properly. Also, the key terms of many equations are not defined.

6. Technical discussion on results is not given. Moreover, the results are not convincing.

7. What is the use Fig. 12? It is not properly visible.

7. The English language is very poor.

8. The organization of the paper is poor. For example, the authors should delete the Study Limitations section. The limitations can be mentioned in the last section. The auhors must go through some reputed journal papers before addressing these comments.

9. Important references are missing, and all the details of the references are not given.

10. Don't include the designation of the auhtor.

Reviewer #3: The article was reviewed under the title " A Hybrid CNN-XLSTM-GRU Deep Learning Model with Autoencoder-Based Feature Selection for Hypothyroidism Diagnosis ".

Overall, the focus of the study is good. However, I would like to offer several recommendations that authors may find useful in the process of revising their manuscript:

1- Based on Figure 1 and the text of the article, it is stated that in the data preparation section, missing values were filled with mean and mode. Was there a specific reason for this or was it done simply based on the similarity of the data?

2- The order of some figures does not match the text of the article in terms of pages, which requires sorting the images for better coherence of the article.

3- In the table related to hyperparameters, the value of epochs is declared as 50, but in the text of the article and in the section Performance outcomes with deep learning models, this value is stated as 200, and even in Figure 11 it is stated as 250. Please review these discrepancies.

4- Regarding comment number 3, it is not clear whether the model was tested in different epochs or whether the early stopping method was used.

5- The resolution of Figures 8 and 10 is low in terms of row and column headings, so it is better to use higher quality figures.

6- Also, the resolution of Figure 12 is low. Considering that it is stated in the section “Error Analysis and Model Robustness” that “Furthermore, Figure 12 shows that the FusionNet-CXG model demonstrated significant…….”, it is better for the resolution of the figure to be higher.

Reviewer #4: The paper is well-written with good quality to meet the criteria of publishing after the improvements for research reproducibility. The authors should add the pseudocode for the proposed approach and the statistical analysis for the results obtained. I recommend Rcommander in Rstudio as a helper tool for that.

6. PLOS authors have the option to publish the peer review history of their article (what does this mean?). If published, this will include your full peer review and any attached files.). If published, this will include your full peer review and any attached files.). If published, this will include your full peer review and any attached files.). If published, this will include your full peer review and any attached files.

...

Reviewer #1: No

Reviewer #2: No

Reviewer #3: No

Reviewer #4: No

---

## [Author Response · Author response to Decision Letter 1]

15 Oct 2025

The authors sincerely thank the Editor and Reviewers for their valuable comments and suggestions, which have helped us to improve the quality and clarity of the manuscript.

---

## [Decision Letter · Decision Letter 1]

11 Nov 2025

PONE-D-25-37245R1A Hybrid CNN-XLSTM-GRU Deep Learning Model with Autoencoder-Based Feature Selection for Hypothyroidism DiagnosisPLOS ONE

Dear Dr. R,

Thank you for submitting your manuscript to PLOS ONE. It was analyzed by four reviewers including me as an Academic Editor (Reviewer #5). After careful consideration, we feel that it has merit but does not fully meet PLOS ONE’s publication criteria as it currently stands. Therefore, we invite you to submit a revised version of the manuscript that addresses the points raised during the review process. **In particular:** 

**the quality of the images should be improved,**

**multiple language problems need to be removed,**

**presentation problems (including imprecise and missing references) must be fixed.**

We look forward to receiving your revised manuscript.

Kind regards,

Maciej Huk, Ph.D.

Academic Editor

PLOS ONE

**Journal Requirements:**

Reviewers' comments:

Reviewer's Responses to Questions

**Comments to the Author**

1. If the authors have adequately addressed your comments raised in a previous round of review and you feel that this manuscript is now acceptable for publication, you may indicate that here to bypass the “Comments to the Author” section, enter your conflict of interest statement in the “Confidential to Editor” section, and submit your "Accept" recommendation.

Reviewer #2: (No Response)

Reviewer #3: (No Response)

Reviewer #4: All comments have been addressed

Reviewer #5: (No Response)

2. Is the manuscript technically sound, and do the data support the conclusions?

Reviewer #2: (No Response)

Reviewer #3: (No Response)

Reviewer #4: Yes

Reviewer #5: Partly

3. Has the statistical analysis been performed appropriately and rigorously? 

Reviewer #2: (No Response)

Reviewer #3: (No Response)

Reviewer #4: Yes

Reviewer #5: I Don't Know

4. Have the authors made all data underlying the findings in their manuscript fully available?

The PLOS Data policy requires authors to make all data underlying the findings described in their manuscript fully available without restriction, with rare exception (please refer to the Data Availability Statement in the manuscript PDF file). The data should be provided as part of the manuscript or its supporting information, or deposited to a public repository. For example, in addition to summary statistics, the data points behind means, medians and variance measures should be available. If there are restrictions on publicly sharing data—e.g. participant privacy or use of data from a third party—those must be specified.requires authors to make all data underlying the findings described in their manuscript fully available without restriction, with rare exception (please refer to the Data Availability Statement in the manuscript PDF file). The data should be provided as part of the manuscript or its supporting information, or deposited to a public repository. For example, in addition to summary statistics, the data points behind means, medians and variance measures should be available. If there are restrictions on publicly sharing data—e.g. participant privacy or use of data from a third party—those must be specified.requires authors to make all data underlying the findings described in their manuscript fully available without restriction, with rare exception (please refer to the Data Availability Statement in the manuscript PDF file). The data should be provided as part of the manuscript or its supporting information, or deposited to a public repository. For example, in addition to summary statistics, the data points behind means, medians and variance measures should be available. If there are restrictions on publicly sharing data—e.g. participant privacy or use of data from a third party—those must be specified.requires authors to make all data underlying the findings described in their manuscript fully available without restriction, with rare exception (please refer to the Data Availability Statement in the manuscript PDF file). The data should be provided as part of the manuscript or its supporting information, or deposited to a public repository. For example, in addition to summary statistics, the data points behind means, medians and variance measures should be available. If there are restrictions on publicly sharing data—e.g. participant privacy or use of data from a third party—those must be specified.

Reviewer #2: (No Response)

Reviewer #3: (No Response)

Reviewer #4: Yes

Reviewer #5: Yes

5. Is the manuscript presented in an intelligible fashion and written in standard English?

Reviewer #2: (No Response)

Reviewer #3: (No Response)

Reviewer #4: Yes

Reviewer #5: No

6. Review Comments to the Author

**Reviewer #2:** The authors have addressed all the previous comments. The quality of this manuscript has been improved.The authors have addressed all the previous comments. The quality of this manuscript has been improved.The authors have addressed all the previous comments. The quality of this manuscript has been improved.The authors have addressed all the previous comments. The quality of this manuscript has been improved.

**Reviewer #3:** (No Response) (No Response) (No Response) (No Response)

**Reviewer #4:** The authors addressed all the comments, I have only one concern about the pesudocode format and the authors should fix it in the final version.The authors addressed all the comments, I have only one concern about the pesudocode format and the authors should fix it in the final version.The authors addressed all the comments, I have only one concern about the pesudocode format and the authors should fix it in the final version.The authors addressed all the comments, I have only one concern about the pesudocode format and the authors should fix it in the final version.

**Reviewer #5:** >>> 1. Language problems:>>> 1. Language problems:>>> 1. Language problems:>>> 1. Language problems:

1.1 European journal of haematology => European Journal of Haematology

1.2 Ieee Access => IEEE Access

1.3 crc Press => CRC Press

1.4 Journal of statistical software => Journal of Statistical Software

>>> 2. Presentation problems:

2.1 "through the use of the rstatix packages [67].": font and its size should be uniform within the text

2.2 Table 3, Table 4: font and its size should be uniform within the text

2.3 Table 5: column 1 - title of the column is missing

2.4 Table 6: table is too big, too much empty space wasted inside the table. Please reformat it to be more compact.

2.5 Invalid reference [90]: "!!! INVALID CITATION !!! (84)."

2.6 Invalid reference [92]: "!!! INVALID CITATION !!! (86, 87)."

2.7 Invalid URL link in reference [58]: "http://crancerminlipigoid/web/packages/caret/vignettes/caretSelectionpdf."

2.8 Not complete reference [59]: "Berrar D. Cross-Validation. 2019."

2.9 Reference [16]: Do not use upercase-only text

2.10 Fig 1. "Disagreement measure" block: the beginning of incomming arrow is positioned imprecisely

2.11 Fig 7: vertical axis: unit is missing (it is not clear if 0.75 is 75% or 0.75%). Measurement error whiskers are missing.

2.12 Reference [67] is imprecise

>>> 3. Other problems: not detected

>>> Recomendation: major rework

===EOT===

7. PLOS authors have the option to publish the peer review history of their article (what does this mean?). If published, this will include your full peer review and any attached files.). If published, this will include your full peer review and any attached files.). If published, this will include your full peer review and any attached files.). If published, this will include your full peer review and any attached files.

...

Reviewer #2: No

Reviewer #3: No

Reviewer #4: No

Reviewer #5: No

---

## [Author Response · Author response to Decision Letter 2]

13 Nov 2025

Reviewer #5: The comments listed (language corrections, table formatting issues, figure inconsistencies, and references related to haematology, rstatix packages, caretSelection, etc.) do not correspond to the content of our manuscript. Our submission does not contain the sections, tables, figures, or references mentioned (e.g., European Journal of Haematology, CRC Press, rstatix, Tables 3–6, Figures 1 and 7, or references [58], [59], [67], [90], [92]). These comments appear to relate to a different manuscript.

---

## [Decision Letter · Decision Letter 2]

5 Jan 2026

PONE-D-25-37245R2A Hybrid CNN-XLSTM-GRU Deep Learning Model with Autoencoder-Based Feature Selection for Hypothyroidism DiagnosisPLOS One

Dear Dr. R,

Thank you for submitting your manuscript to PLOS ONE. It was analyzed by two reviewers including me as an Academic Editor (Reviewer #5). After careful consideration, we feel that it has merit but does not fully meet PLOS ONE’s publication criteria as it currently stands. Therefore, we invite you to submit a revised version of the manuscript that addresses the points raised during the review process.

In particular: 

the quality of some of the images should be improved (especially Fig. 6),multiple presentation problems must be fixed.

We look forward to receiving your revised manuscript.

Kind regards,

Maciej Huk, Ph.D.

Academic Editor

PLOS One

Journal Requirements:

Reviewers' comments:

Reviewer's Responses to Questions

**Comments to the Author**

1. If the authors have adequately addressed your comments raised in a previous round of review and you feel that this manuscript is now acceptable for publication, you may indicate that here to bypass the “Comments to the Author” section, enter your conflict of interest statement in the “Confidential to Editor” section, and submit your "Accept" recommendation.

Reviewer #3: (No Response)

Reviewer #5: (No Response)

2. Is the manuscript technically sound, and do the data support the conclusions?

Reviewer #3: (No Response)

Reviewer #5: Partly

3. Has the statistical analysis been performed appropriately and rigorously? 

Reviewer #3: (No Response)

Reviewer #5: I Don't Know

4. Have the authors made all data underlying the findings in their manuscript fully available?

The PLOS Data policy requires authors to make all data underlying the findings described in their manuscript fully available without restriction, with rare exception (please refer to the Data Availability Statement in the manuscript PDF file). The data should be provided as part of the manuscript or its supporting information, or deposited to a public repository. For example, in addition to summary statistics, the data points behind means, medians and variance measures should be available. If there are restrictions on publicly sharing data—e.g. participant privacy or use of data from a third party—those must be specified.requires authors to make all data underlying the findings described in their manuscript fully available without restriction, with rare exception (please refer to the Data Availability Statement in the manuscript PDF file). The data should be provided as part of the manuscript or its supporting information, or deposited to a public repository. For example, in addition to summary statistics, the data points behind means, medians and variance measures should be available. If there are restrictions on publicly sharing data—e.g. participant privacy or use of data from a third party—those must be specified.requires authors to make all data underlying the findings described in their manuscript fully available without restriction, with rare exception (please refer to the Data Availability Statement in the manuscript PDF file). The data should be provided as part of the manuscript or its supporting information, or deposited to a public repository. For example, in addition to summary statistics, the data points behind means, medians and variance measures should be available. If there are restrictions on publicly sharing data—e.g. participant privacy or use of data from a third party—those must be specified.requires authors to make all data underlying the findings described in their manuscript fully available without restriction, with rare exception (please refer to the Data Availability Statement in the manuscript PDF file). The data should be provided as part of the manuscript or its supporting information, or deposited to a public repository. For example, in addition to summary statistics, the data points behind means, medians and variance measures should be available. If there are restrictions on publicly sharing data—e.g. participant privacy or use of data from a third party—those must be specified.

Reviewer #3: (No Response)

Reviewer #5: Yes

5. Is the manuscript presented in an intelligible fashion and written in standard English?

Reviewer #3: (No Response)

Reviewer #5: Yes

6. Review Comments to the Author

Reviewer #3: The article titled " A Hybrid CNN-XLSTM-GRU Deep Learning Model with Autoencoder-Based Feature

Selection for Hypothyroidism Diagnosis " was reviewed again.

This revision has significantly improved the manuscript. The authors have satisfactorily addressed my concerns.

Reviewer #5: >>> 1. Language problems: not detected

>>> 2. Preseentation problems:

2.1 Fig 6.: values are unreadable. This should not happen.

Presentation of such figure has very limited value, and decreases the overall quality of the manuscript.

2.2 Table 5 repeates multiple information from Table 4. Is this needed?

2.3 Table 5 includes elements such as "Storage: Google Drive", "Operating System: Google Colab Environment"

Are results influenced by those elements (e.g by the type of data storage)? If yes it should be discussed. If no, then not needed information should be removed.

Btw. Google Colab Environment is not an Operating System (typically it uses some form of Linux as an OS).

2.4 Fig 9, Fig 10b, Fig 11, Fig 12b, Fig 13, Fig 14b, Fig 15: vertical axis, description of the axis should present units [% or 1? it is not clear if 0.918 means 0.918% or 91,8%]

2.5 Fig 9, Fig 11, Fig 13, Fig 15, Fig 16, Table 9, Table 10 : title is not precise; it is not written if presented result is for traiing or testing data.

2.6 2.5 Fig 9, Fig 11, Fig 13, Fig 15: Error whiskers or regions should be also presented withn those figures.

2.7 Table 8: "Â" character before "±" seems to be improper (all rows including the header)

2.8 Fig 2, Fig 4, Fig 15- Fig 17: There is no need to repeat title above the figure. Title below the figure should be enough.

2.9 Table 11. It would be more clear if "Note: Entries marked with (†) correspond to studies that used datasets different from the UCI/Kaggle hypothyroid dataset employed in this work." was placed just under the table (e.g. as a footnote) or within its title.

>>> 3. Other problems:

3.1 Performed comparison of various models is not based on statistical tests. This limits the confidence of presented analysis and conclusions. It should be discussed as an limitation of the study.

>>> Recommendation: Major revision

>>> Additional comment: My previous review uploaded for PONE-D-25-37245R1 was related with different manuscript. The review provided above is related with PONE-D-25-37245R2.

=== EOT ===

7. PLOS authors have the option to publish the peer review history of their article (what does this mean?). If published, this will include your full peer review and any attached files.). If published, this will include your full peer review and any attached files.). If published, this will include your full peer review and any attached files.). If published, this will include your full peer review and any attached files.

...

Reviewer #3: No

Reviewer #5: No

---

## [Author Response · Author response to Decision Letter 3]

13 Jan 2026

The authors thank the reviewers for their valuable comments and suggestions to improve the quality of the manuscript.

---

## [Decision Letter · Decision Letter 3]

8 Mar 2026

PONE-D-25-37245R3A Hybrid CNN-XLSTM-GRU Deep Learning Model with Autoencoder-Based Feature Selection for Hypothyroidism DiagnosisPLOS One

Dear Dr. R,

Thank you for submitting your manuscript to PLOS ONE. It was additionally verified by me acting as an Academic Editor (Reviewer #5). After careful consideration, we feel that it has merit but does not fully meet PLOS ONE’s publication criteria as it currently stands. Therefore, we invite you to submit a revised version of the manuscript that addresses the points raised during the review process. In particular:

language and presentation problems should be removed,missing details of statistical analysis should be clarified.

We look forward to receiving your revised manuscript.

Kind regards,

Maciej Huk, Ph.D.

Academic Editor

PLOS One

Journal Requirements:

Reviewers' comments:

Reviewer's Responses to Questions

**Comments to the Author**

1. If the authors have adequately addressed your comments raised in a previous round of review and you feel that this manuscript is now acceptable for publication, you may indicate that here to bypass the “Comments to the Author” section, enter your conflict of interest statement in the “Confidential to Editor” section, and submit your "Accept" recommendation.

Reviewer #5: (No Response)

2. Is the manuscript technically sound, and do the data support the conclusions?

Reviewer #5: Partly

3. Has the statistical analysis been performed appropriately and rigorously? 

Reviewer #5: I Don't Know

4. Have the authors made all data underlying the findings in their manuscript fully available?

The PLOS Data policy requires authors to make all data underlying the findings described in their manuscript fully available without restriction, with rare exception (please refer to the Data Availability Statement in the manuscript PDF file). The data should be provided as part of the manuscript or its supporting information, or deposited to a public repository. For example, in addition to summary statistics, the data points behind means, medians and variance measures should be available. If there are restrictions on publicly sharing data—e.g. participant privacy or use of data from a third party—those must be specified.requires authors to make all data underlying the findings described in their manuscript fully available without restriction, with rare exception (please refer to the Data Availability Statement in the manuscript PDF file). The data should be provided as part of the manuscript or its supporting information, or deposited to a public repository. For example, in addition to summary statistics, the data points behind means, medians and variance measures should be available. If there are restrictions on publicly sharing data—e.g. participant privacy or use of data from a third party—those must be specified.requires authors to make all data underlying the findings described in their manuscript fully available without restriction, with rare exception (please refer to the Data Availability Statement in the manuscript PDF file). The data should be provided as part of the manuscript or its supporting information, or deposited to a public repository. For example, in addition to summary statistics, the data points behind means, medians and variance measures should be available. If there are restrictions on publicly sharing data—e.g. participant privacy or use of data from a third party—those must be specified.requires authors to make all data underlying the findings described in their manuscript fully available without restriction, with rare exception (please refer to the Data Availability Statement in the manuscript PDF file). The data should be provided as part of the manuscript or its supporting information, or deposited to a public repository. For example, in addition to summary statistics, the data points behind means, medians and variance measures should be available. If there are restrictions on publicly sharing data—e.g. participant privacy or use of data from a third party—those must be specified.

Reviewer #5: Yes

5. Is the manuscript presented in an intelligible fashion and written in standard English?

Reviewer #5: Yes

6. Review Comments to the Author

Reviewer #5: >>> 1. Language problems:

1.1 "Precision refers to all positively labelled items. recall refers to a positive(hypothyroid) class forecast."

=> "Precision refers to all positively labelled items. Recall refers to a positive(hypothyroid) class forecast."

1.2 "The Mathematical formulae for various statistical methods used in model assessment are delineated in Eqs (15-20)."

=> "The mathematical formulae for various statistical methods used in model assessment are delineated in Eqs (15-20)."

>>> 2. Presentation problems:

2.1 Fig 2: vertical axis: are Authors sure that "Frequency" is the right description. Please consider "Number of subjects" or "Count"

2.2 Fig 15 - blue blocks are not translucent and hide other information

2.3 Fig 10b, Fig 12b, Fig 14b: Vertical axis (accuracy) should present units (0.9 can mean 90% as well as 0.9%)

2.4 Fig 9, Fig 11, Fig 13: axis desxriptions are of low quality. Please consider changing figure to vector form.

Also the legend entry "95% CI whiskers" is redundant (and not needed).

2.5 Fig 16: The title is too general. Is it done on training or testing data

Also: Error whiskers overlap with some values (this limits readability). Please move the values.

2.6 Table 9: estimated measurement errors should be presented

2.7 numbering of equations: (8), (14) are improperly justified (should be: to the right)

>>> 3. Other problems:

3.1 Table 10: Authors are using paired t-test. It is not clear if t-test assumptions were verified (distributions normality and similarity of variances)?

3.2 It is not clear what 5x10 fold CV means: is it 10 times repeated 5-fold CV or 5 times repeated 10-fold CV?

3.3 It is not clear how the t-test was performed. Treating separate folds as data blocks in statistical test is not a good solution as folds are statistically not independent. Folds should be averaged and repeated CV should be used to multiply data blocks.

3.4 It is not clear how averaging of classification accuracy was done - was it micro or macro averaging?

>>> Recommendation: major revision

===EOT===

7. PLOS authors have the option to publish the peer review history of their article (what does this mean?). If published, this will include your full peer review and any attached files.). If published, this will include your full peer review and any attached files.). If published, this will include your full peer review and any attached files.). If published, this will include your full peer review and any attached files.

...

Reviewer #5: No

---

## [Author Response · Author response to Decision Letter 4]

16 Mar 2026

The authors thank the reviewer for this valuable comment. We have revised the manuscript accordingly.

---

## [Decision Letter · Decision Letter 4]

21 Mar 2026

PONE-D-25-37245R4A Hybrid CNN-XLSTM-GRU Deep Learning Model with Autoencoder-Based Feature Selection for Hypothyroidism DiagnosisPLOS One

Dear Dr. R,

Thank you for submitting your manuscript to PLOS ONE. It was additionally verified by me acting as an Academic Editor (Reviewer #5). After careful consideration, we feel that it has merit but does not fully meet PLOS ONE’s publication criteria as it currently stands. Therefore, we invite you to submit a revised version of the manuscript that addresses the points raised during the review process.

In particular:

minor presentation problems should be removed,indicated details of statistical analysis should be clarified.

We look forward to receiving your revised manuscript.

Kind regards,

Maciej Huk, Ph.D.

Academic Editor

PLOS One

Journal Requirements:

Reviewers' comments:

Reviewer's Responses to Questions

**Comments to the Author**

1. If the authors have adequately addressed your comments raised in a previous round of review and you feel that this manuscript is now acceptable for publication, you may indicate that here to bypass the “Comments to the Author” section, enter your conflict of interest statement in the “Confidential to Editor” section, and submit your "Accept" recommendation.

Reviewer #5: (No Response)

2. Is the manuscript technically sound, and do the data support the conclusions?

Reviewer #5: Partly

3. Has the statistical analysis been performed appropriately and rigorously? 

Reviewer #5: I Don't Know

4. Have the authors made all data underlying the findings in their manuscript fully available?

The PLOS Data policy requires authors to make all data underlying the findings described in their manuscript fully available without restriction, with rare exception (please refer to the Data Availability Statement in the manuscript PDF file). The data should be provided as part of the manuscript or its supporting information, or deposited to a public repository. For example, in addition to summary statistics, the data points behind means, medians and variance measures should be available. If there are restrictions on publicly sharing data—e.g. participant privacy or use of data from a third party—those must be specified.requires authors to make all data underlying the findings described in their manuscript fully available without restriction, with rare exception (please refer to the Data Availability Statement in the manuscript PDF file). The data should be provided as part of the manuscript or its supporting information, or deposited to a public repository. For example, in addition to summary statistics, the data points behind means, medians and variance measures should be available. If there are restrictions on publicly sharing data—e.g. participant privacy or use of data from a third party—those must be specified.requires authors to make all data underlying the findings described in their manuscript fully available without restriction, with rare exception (please refer to the Data Availability Statement in the manuscript PDF file). The data should be provided as part of the manuscript or its supporting information, or deposited to a public repository. For example, in addition to summary statistics, the data points behind means, medians and variance measures should be available. If there are restrictions on publicly sharing data—e.g. participant privacy or use of data from a third party—those must be specified.requires authors to make all data underlying the findings described in their manuscript fully available without restriction, with rare exception (please refer to the Data Availability Statement in the manuscript PDF file). The data should be provided as part of the manuscript or its supporting information, or deposited to a public repository. For example, in addition to summary statistics, the data points behind means, medians and variance measures should be available. If there are restrictions on publicly sharing data—e.g. participant privacy or use of data from a third party—those must be specified.

Reviewer #5: Yes

5. Is the manuscript presented in an intelligible fashion and written in standard English?

Reviewer #5: Yes

6. Review Comments to the Author

Reviewer #5:

>>> 1. Language problems: not detected

>>> 2. Presentation problems:

2.1 Fig 15 - deep blue blocks are not translucent and hide other information - please consider making them a little translucent

2.2 Fig 16: The title is not specyfying if presented results were obtained on training or on testing data.

2.3 numbering of equation: (8) is improperly alligned (should be: to the right)

2.4 Autors write: "The f1 score is the average (moderate) of recall and precision"

f1 => F1

2.5 Space is needed between title of section 3.7 and preceeding paragraph

>>> 3. Other problems:

3.1 Table 10: Authors are using paired t-test. It is not clear if assumptions of t-test were verified (normality of distributions of results and similarity of their variances). This should be verified and discussed within the text. If the assumptions are not met then not parametric test should be used instead of t-test.

3.2 Authors write: "These repetition-level mean accuracies were then subjected to the paired ttest thus eliminating the data on individual folds being considered statistically independent observations."

This can suggest that results for folds are "considered statistically independent observations".

Do Authors mean "These repetition-level mean accuracies were then subjected to the paired ttest thus eliminating the data on individual folds being considered statistically dependent observations." ?

>>> Recommendation: minor revision

===EOT===

7. PLOS authors have the option to publish the peer review history of their article (what does this mean?). If published, this will include your full peer review and any attached files.). If published, this will include your full peer review and any attached files.). If published, this will include your full peer review and any attached files.). If published, this will include your full peer review and any attached files.

...

Reviewer #5: No

---

## [Author Response · Author response to Decision Letter 5]

24 Mar 2026

Thank you to the Editor and Reviewers for their valuable comments. All comments have been addressed carefully, and the manuscript has been revised accordingly. The revisions improved the clarity, methodology, statistical explanation, presentation, and language of the paper.

A detailed point-by-point response to all editor and reviewer comments has been prepared and submitted. We believe the revised manuscript satisfactorily addresses the concerns raised.

We sincerely thank the Editor and Reviewers for their time and helpful suggestions.

---

## [Decision Letter · Decision Letter 5]

30 Mar 2026

A Hybrid CNN-XLSTM-GRU Deep Learning Model with Autoencoder-Based Feature Selection for Hypothyroidism Diagnosis

PONE-D-25-37245R5

Dear Dr. R,

Thank you for submitting your manuscript to PLOS ONE. It was additionally verified by me acting as an Academic Editor (Reviewer #5). We’re pleased to inform you that your manuscript has been judged scientifically suitable for publication and will be formally accepted for publication once it meets all outstanding technical requirements.

Kind regards,

Maciej Huk, Ph.D.

Academic Editor

PLOS One

Additional Editor Comments (optional):

Reviewers' comments:

Reviewer's Responses to Questions

**Comments to the Author**

1. If the authors have adequately addressed your comments raised in a previous round of review and you feel that this manuscript is now acceptable for publication, you may indicate that here to bypass the “Comments to the Author” section, enter your conflict of interest statement in the “Confidential to Editor” section, and submit your "Accept" recommendation.

Reviewer #5: All comments have been addressed

2. Is the manuscript technically sound, and do the data support the conclusions?

Reviewer #5: Yes

3. Has the statistical analysis been performed appropriately and rigorously? 

Reviewer #5: I Don't Know

4. Have the authors made all data underlying the findings in their manuscript fully available?

The PLOS Data policy requires authors to make all data underlying the findings described in their manuscript fully available without restriction, with rare exception (please refer to the Data Availability Statement in the manuscript PDF file). The data should be provided as part of the manuscript or its supporting information, or deposited to a public repository. For example, in addition to summary statistics, the data points behind means, medians and variance measures should be available. If there are restrictions on publicly sharing data—e.g. participant privacy or use of data from a third party—those must be specified.requires authors to make all data underlying the findings described in their manuscript fully available without restriction, with rare exception (please refer to the Data Availability Statement in the manuscript PDF file). The data should be provided as part of the manuscript or its supporting information, or deposited to a public repository. For example, in addition to summary statistics, the data points behind means, medians and variance measures should be available. If there are restrictions on publicly sharing data—e.g. participant privacy or use of data from a third party—those must be specified.requires authors to make all data underlying the findings described in their manuscript fully available without restriction, with rare exception (please refer to the Data Availability Statement in the manuscript PDF file). The data should be provided as part of the manuscript or its supporting information, or deposited to a public repository. For example, in addition to summary statistics, the data points behind means, medians and variance measures should be available. If there are restrictions on publicly sharing data—e.g. participant privacy or use of data from a third party—those must be specified.requires authors to make all data underlying the findings described in their manuscript fully available without restriction, with rare exception (please refer to the Data Availability Statement in the manuscript PDF file). The data should be provided as part of the manuscript or its supporting information, or deposited to a public repository. For example, in addition to summary statistics, the data points behind means, medians and variance measures should be available. If there are restrictions on publicly sharing data—e.g. participant privacy or use of data from a third party—those must be specified.

Reviewer #5: Yes

5. Is the manuscript presented in an intelligible fashion and written in standard English?

Reviewer #5: Yes

6. Review Comments to the Author

Reviewer #5: Authors have fixed all earlier detected problems. The text can be regarded as ready for publication.

>>> 1. Language problems: not detected

>>> 2. Presentation problems: not detected

>>> 3. Other problems: not detected

>>> Recommendation: Accept

7. PLOS authors have the option to publish the peer review history of their article (what does this mean?). If published, this will include your full peer review and any attached files.). If published, this will include your full peer review and any attached files.). If published, this will include your full peer review and any attached files.). If published, this will include your full peer review and any attached files.

...

Reviewer #5: No

---

## [Editor Report · Acceptance letter]

PONE-D-25-37245R5

PLOS One

Dear Dr. R,

I'm pleased to inform you that your manuscript has been deemed suitable for publication in PLOS One. Congratulations! Your manuscript is now being handed over to our production team.

Kind regards,

on behalf of

Dr. Maciej Huk

Academic Editor

PLOS One